# Bst2-targeted senotherapy restores visual function by eliminating senescent retinal cells

Jun Yong Oh[1,8], Jae-Byoung Chae[2,8], Hyo Kyung Lee[3], Chul-Woo Park[2], Minseo Bae[2], Gyuri Kim[4], Yujeong Oh[4], Gyeongseok Yang[1,5], Sangpil Kim[1], Hae Won Ok[1], Dohyun Kim[1], Chaekyu Kim[6], Semin Lee[3], Jiwon Jang[4], Hyewon Chung[2,7] ✉ & Ja-Hyoung Ryu[1,5] ✉

Senescent cells contribute to degenerative processes in multiple tissues, including the retina. In the retinal pigment epithelium (RPE), their accumulation is closely associated with retinal aging and disease progression. Eliminating senescent RPE cells has shown therapeutic potential, but conventional senolytics often lack the specificity required to spare non-senescent cells, raising safety concerns. To overcome this, we performed integrated transcriptomic analyses of male mouse-derived RPE cells under natural aging and chemically induced senescence conditions. These analyses identified Bst2 as a membrane-localized marker selectively upregulated in senescent RPE cells, with minimal expression in young controls. Based on this discovery, we developed a modular, antibody-pluggable drug delivery platform–B-Z-PON–comprising mesoporous silica nanoparticles functionalized with a recombinant Fc-binding domain and conjugated with anti-Bst2 antibodies. This nanocarrier selectively accumulates in Bst2-expressing senescent RPE cells, enabling targeted drug delivery and sparing healthy retinal cells. In vivo administration of ABT-263-loaded B-Z-PON in aged and senescence-induced retinal degeneration models resulted in the selective ablation of senescent cells, restoration of RPE function, and improved visual outcomes. Together, our study integrates senescence-specific marker discovery with precision nanomedicine, establishing a versatile platform for targeted senotherapy. These findings offer a promising therapeutic approach for retinal aging disorders, such as age-related macular degeneration.

Ageing is a systemic and multifactorial process marked by the progressive accumulation of cellular and molecular damage, ultimately contributing to tissue dysfunction and chronic disease across multiple organ systems[1]. Among the hallmarks of ageing, cellular senescence–a state of irreversible cell-cycle arrest accompanied by a proinflammatory secretory phenotype (senescence-associated secretory phenotype, SASP)–has emerged as a major driver of age-associated degeneration in diverse tissues including musculoskeletal, cardiovascular, and nervous system[1–4]. Senotherapeutic approaches aimed at eliminating senescent cells (senolytics) or modulating their secretory phenotype (senomorphics) have shown promise in extending healthspan and improving organ function in preclinical models[5–9].

The retina, particularly the retinal pigment epithelium (RPE)–a monolayer of terminally differentiated cells essential for

photoreceptor survival and visual function–is especially vulnerable to age-related decline due to its high metabolic activity and limited regenerative capacity[10,11]. Emerging evidence implicates RPE senescence as a critical upstream pathology in age-related macular degeneration (AMD), a leading cause of irreversible vision loss in the elderly population[12–14]. Although the RPE has only limited proliferative capacity under physiological conditions, it exhibits hallmark features of cellular senescence in response to cumulative oxidative, metabolic, and environmental stress[15–17]. Such changes have been observed in both dry AMD, characterized by RPE and photoreceptor atrophy, and neovascular AMD, defined by choroidal neovascularization[13,15].

These findings have prompted interest in senescence-targeted therapies as a strategy to halt or reverse retinal degeneration. Among them, senolytics hold particular promise, but their clinical translation has been hindered by key limitations. First-generation compounds such as dasatinib, quercetin, and fisetin have shown efficacy in various models, yet often suffer from poor pharmacokinetics, systemic toxicity, and a lack of tissue specificity[18–20]–significant concerns in the retina, a highly delicate and vision-critical structure. Even the local delivery, precise targeting is required to minimize off-target effects and preserve retinal integrity. In our previous studies, the BCL-2 inhibitor ABT-263 (navitoclax) and the MDM2 antagonist Nutlin-3a effectively cleared senescent RPE cells and ameliorated retinal pathology[21–23]. However, broader therapeutic application remains limited by delivery challenges and safety concerns.

Nanomedicine offers a promising solution to these limitations[24]. Nanoparticles provide a modular platform for drug encapsulation, controlled release, and surface engineering, improving pharmacokinetics and enabling cell-selective targeting[25,26]. Importantly, senescence-responsive nanocarriers can be engineered to release their payload in response to hallmarks of senescence, such as elevated β-galactosidase activity or altered redox homeostasis, allowing selective action in senescent cells while sparing healthy tissue[27]. These approaches have improved therapeutic efficacy and reduced toxicity in pre-clinical models. Nonetheless, achieving true precision senotherapy requires coupling nanocarriers with senescence-specific surface markers capable of guiding targeted delivery to pathogenic senescent cells[28]. Notably, early human studies support the clinical plausibility of senolytic interventions, including intermittent dasatinib plus quercetin treatment in individuals with diabetic kidney disease[29]. In ophthalmology, a recent preclinical-to-phase 1 study reported initial clinical translation of senolytic targeting in diabetic macular edema[30], underscoring both the relevance of senescence as a therapeutic target and the need for improved tissue/phenotype selectivity in the eye.

Motivated by this emerging clinical translation and the need for improved tissue selectivity, in this study, we identify Bone Marrow Stromal Cell Antigen 2 (Bst2; CD317/tetherin) as a senescence-associated surface marker enriched in senescent RPE cells. Bst2 is a type II transmembrane protein best known as an interferon-inducible antiviral effector that can be upregulated under cellular stress and innate immune activation[31,32]. To define surface candidates linked to RPE ageing and senescence, we reanalyzed two single-cell RNA-seq datasets from our previous studies capturing natural ageing (young vs aged mouse RPE) and doxorubicin (Dox)-induced injury with senescence-like features in mouse RPE[33,34]. In both contexts, Bst2 showed selective enrichment within a senescent RPE cluster, supporting its utility as a practical targeting moiety for cell-selective senotherapeutic delivery in the retina.

To translate this discovery into a therapeutic platform, we developed a Bst2-targeted nanoparticle system for selective drug delivery. We engineered antibody-binding domain-conjugated protein nanoparticles (Z-PON), functionalized with a recombinant GST-ABD protein capable of non-covalently binding the Fc region of anti-Bst2 antibodies[35,36]. This modular design enabled antibody attachment via protein-protein interaction without the need for chemical

crosslinking[37]. The resulting Bst2-targeted nanoparticles (B-Z-PON) demonstrated selective accumulation in senescent RPE cells and effective clearance in vitro and in vivo. Together, our findings establish Bst2 as a robust biomarker of RPE senescence and lay the foundation for targeted senolytic therapies in age-related retinal degeneration.

## Results

### Bst2 emerges as a selective senescence marker in aging RPE cells

To identify cell-surface markers associated with RPE aging and senescence, we performed a comparative analysis of two independent single-cell RNA sequencing (scRNA-seq) datasets representing (i) natural aging and (ii) Dox-induced senescence-like injury in mouse RPE. To profile transcriptional changes during natural aging in RPE, we analyzed our previously generated dataset (GSE282283)[33]. After quality control and clustering, we identified major RPE/choroidal cell populations by UMAP and extracted 7908 RPE cells, including 2597 young-RPE (3 months) and 5311 aged-RPE (24 months) cells (Fig. 1a). Differential gene expression analysis (Seurat FindMarkers; Wilcoxon rank-sum test) identified 21 upregulated genes in aged-RPE relative to young-RPE (log₂Fold Change (FC) > 0.5, adjusted $P$ value < 0.05). To validate conserved candidates in an independent context, we reanalyzed a scRNA-seq dataset of mouse RPE/choroid following subretinal Dox injection (GSE183572)[34], a model previously validated for RPE senescence and senolytic evaluation[21,38,39]. We extracted 3319 RPE cells comprising 1760 Dox-injected RPE (Dox-RPE) and 1559 control RPE cells (Fig. 1b). Differential expression analysis identified 9 upregulated genes in Dox-RPE relative to controls (log₂FC > 0.5, adjusted $P$ < 0.05). To identify candidates shared between the two models, we intersected the upregulated gene sets (Fig. 1c), yielding two overlapping genes: Bst2, a membrane-localized protein[31,32], and Ifi27, an interferon-inducible pro-apoptotic factor[40]. Notably, Bst2 upregulation was most prominent within the RPE population in the natural aging dataset (log₂FC = 0.503, adjusted $P$ value < 0.05), whereas adjacent non-RPE populations (e.g., choroidal fibroblasts) did not display comparable upregulation (Fig. 1d, log₂FC = 0.35). Consistently, Bst2 transcript levels were higher in both aged-RPE vs young-RPE and Dox-RPE vs control RPE (Fig. 1e, $t$ test: $P$ value < 0.0001). At the protein level, immunoblotting of RPE tissue from Dox-injected mice and aged mice confirmed increased Bst2 expression alongside established senescence markers p53 and p21 (Fig. 1f, g). Immunofluorescence staining for p53 or p21 showed nuclear localization of p53 or p21 in the RPE and increased punctate Bst2 signals in the aged RPE layer (Fig. 1h and Supplementary Fig. 1). Bst2⁺p53⁺ and Bst2⁺p21⁺ cells were rarely detected in young RPE but were significantly enriched in aged RPE, as quantified by the number of double-positive cells per field. Together, these data position Bst2 as a candidate RPE-enriched surface marker associated with aging and senescence-like states, providing a rational basis for targeted senotherapeutic delivery in subsequent experiments.

### Bst2 knockdown does not attenuate Dox-induced senescence in RPE cells

To evaluate whether Bst2 plays a functional role in RPE senescence, we performed loss-of-function experiments in the widely used human RPE cell line ARPE-19. We used a previously established in vitro Dox-induced senescence model in ARPE-19 cells[21,22,39] and revalidated senescence induction in the current study using orthogonal criteria, including SA-β-gal activity, durable proliferative arrest after drug washout, persistent DNA-damage signaling, and expression of key senescence markers (Supplementary Fig. 2). ARPE-19 cells were transduced with lentiviral vectors encoding either shScramble or Bst2-targeting shRNA (shBst2) and selected with puromycin, followed by vehicle or Dox treatment to induce senescence (Fig. 2a). Under basal conditions, Bst2 signals were low in both groups (Fig. 2b). After Dox treatment, Bst2 expression was robustly induced in shScramble cells

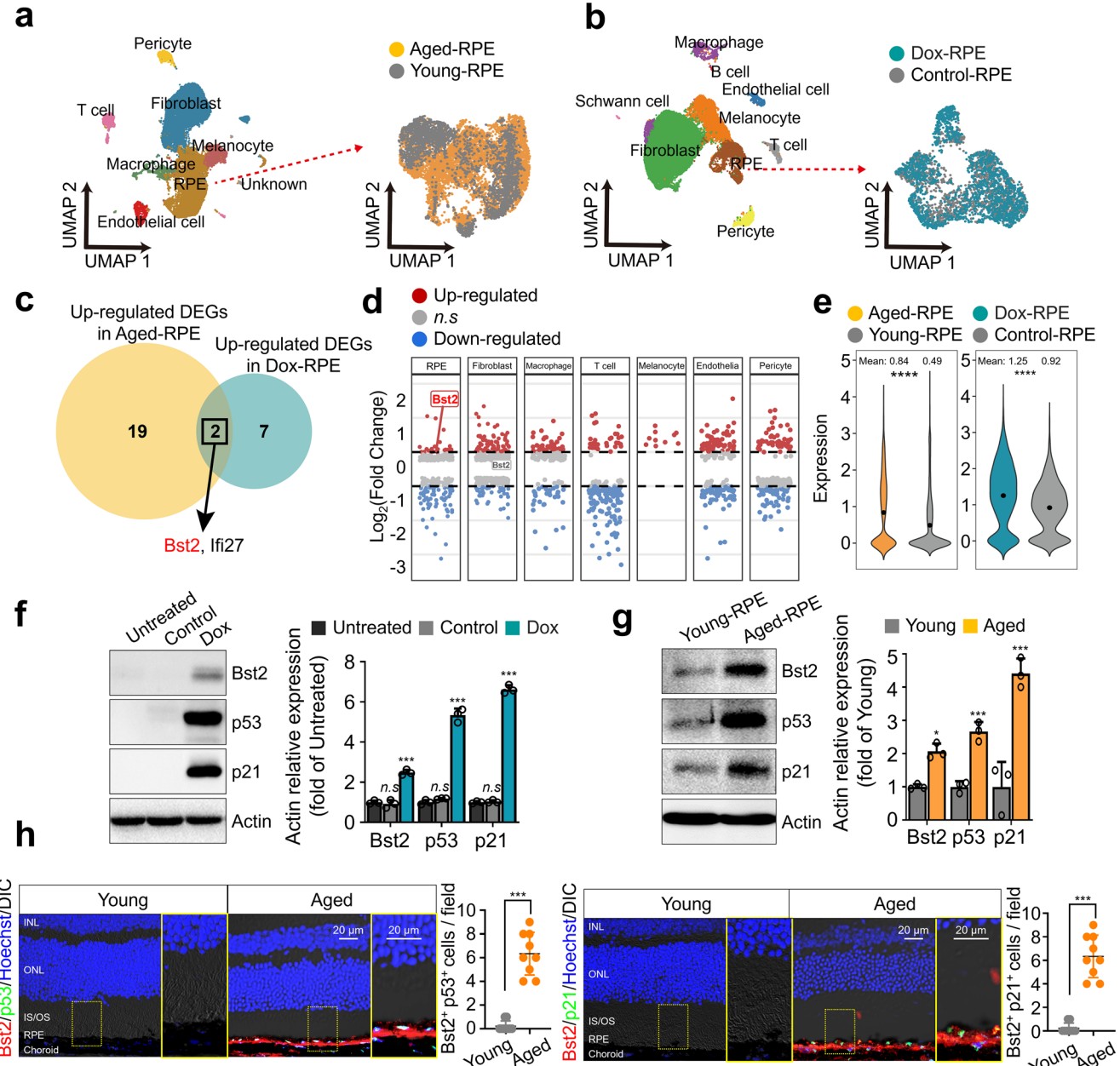

**Fig. 1 | Identification of a selective RPE senescence marker.** UMAP of RPE/choroid cells from young vs. naturally aged mice (GSE282283) (**a**) and from subretinal injection (SRI) vehicle- vs. doxorubicin (Dox)-injected mouse RPE/choroid (GSE183572) (**b**), with the corresponding RPE reclustering shown on the right. **c** Venn diagram showing overlapping upregulated differentially expressed genes (DEGs) in aged RPE (Aged-RPE vs Young-RPE) and Dox-induced senescent RPE (Dox-RPE vs Control-RPE). **d** Cell type-resolved log2 fold-change ($\log_2$FC) of DEGs comparing Aged-RPE to Young-RPE in the naturally aged dataset. Bst2 is selectively upregulated in RPE ($\log_2$FC > 0.5) relative to other cell types. **e** Bst2 expression in the RPE populations from GSE282283 and GSE183572. Immunoblotting of Bst2, p53, and p21 in mouse RPE lysates from the subretinal injection (SRI) model: untreated,

vehicle-injected, and Dox-injected eyes (**f**), and from naturally young (3-month-old) and aged (24-month-old) mice (**g**) ($n = 3$ biological replicates). Quantification of band intensities normalized to β-actin is shown in the right panels. **h** Representative immunofluorescence images of retinal sections from young and aged mice stained for Bst2 with p53 (left) or p21 (right), with Hoechst counterstaining. Yellow boxes indicate regions shown at higher magnification. Quantification plots show the number of Bst2+p53+ cells or Bst2+p21+ cells per field within the RPE layer ($n = 3$ biological replicates). Scale bars, 20 μm. Statistical analyses were performed using one-way ANOVA with Tukey's multiple-comparisons test (**f**) and two-tailed unpaired $t$ test (**e**, **g**, and **h**). Data are presented as mean ± SD. *$P < 0.05$, **$P < 0.01$, ***$P < 0.001$, ****$P < 0.0001$. Source data are provided as a Source Data file.

but effectively suppressed in the shBst2 group (Fig. 2b), confirming efficient knockdown. However, Bst2 knockdown did not attenuate senescence-associated phenotypes: SA-β-gal staining showed no significant difference between shScramble and shBst2 cells under Dox-induced conditions (Fig. 2c). Immunoblotting further confirmed efficient Bst2 suppression, whereas canonical senescence markers (p53, p21, and p16) remained comparable between shScramble and shBst2 groups (Fig. 2d). Consistently, immunofluorescence analyses showed

no significant differences in the proportions of p53-, p21-, or p16-positive cells between groups, and persistent DNA-damage signaling was similar as quantified by γH2AX foci (Fig. 2e).

We next examined whether these findings translate in vivo using a Dox-induced mouse RPE senescence model. In this model, subretinal Dox induces a stable senescence-like state in the RPE by day 7, enabling therapeutic intervention and endpoint analyses at day 10, as previously validated for senescence assessment and senolytic evaluation[21,38,39].

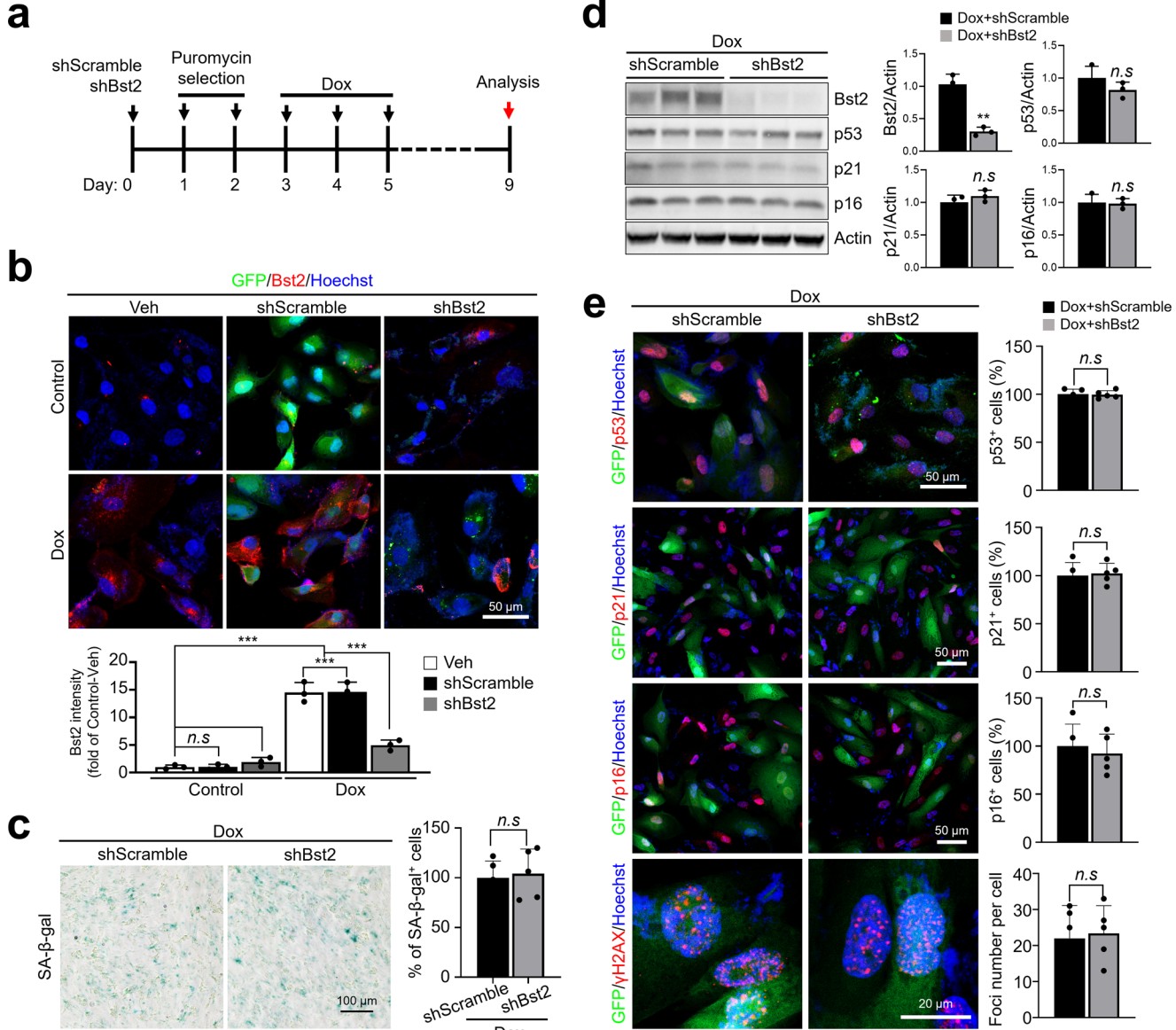

**Fig. 2 | Senescence phenotypes after Bst2 knockdown in Dox-treated RPE cells.** **a** Experimental scheme for shRNA transduction, puromycin selection, Dox treatment, recovery, and endpoint analysis. **b** Representative immunofluorescence images of Bst2 in control (vehicle) and Dox-treated ARPE-19 cells transduced with shScramble or shBst2. GFP marks transduced cells; nuclei were counterstained with Hoechst (blue) ($n = 3$ biological replicates). **c** Representative SA-β-gal staining images and quantification of SA-β-gal–positive cells in Dox-treated shScramble and shBst2 conditions ($n = 5$ biological replicates). **d** Immunoblot analysis confirming Bst2 knockdown and assessing canonical senescence markers (p53, p21, p16) in Dox-treated cells; densitometric quantification (normalized to actin) is shown at right ($n = 3$ biological replicates). **e** Representative immunofluorescence images and quantification of p53, p21, p16, and γH2AX foci in Dox-treated shScramble and shBst2 cells ($n = 5$ biological replicates). For γH2AX, the number of foci per nucleus was quantified and did not differ between groups ($22.1 \pm 9.1$ for shScramble vs $23.4 \pm 7.6$ for shBst2 foci/nucleus). Scale bars: 50 μm in (**b**) and in the p53/p21/p16 panels of (**e**); 100 μm in (**c**); 20 μm in the high-magnification γH2AX panel in (**e**). Statistical analyses were performed using one-way ANOVA followed by Tukey's multiple-comparisons test and two-tailed unpaired $t$ tests, as appropriate. Data are presented as the mean ± SD. \*\*$P < 0.01$ and \*\*\*$P < 0.001$. Source data are provided as a Source Data file.

Subretinal delivery of AAV2/5 encoding shBst2 reduced Bst2 expression in the RPE but did not measurably alter the expression of senescence markers (p53, p21, and p16) following Dox exposure (Fig. 3a, b). Quantification of SA-β-gal–positive area showed robust induction by Dox, with no significant difference between shScramble and shBst2 groups (Fig. 3c). ERG recordings further demonstrated that Dox-induced reductions in the c-wave (RPE function), as well as a- and b-wave amplitudes, were comparable between shScramble and shBst2 groups (Fig. 3d). Together, these data indicate that although Bst2 is consistently upregulated in senescent RPE, its knockdown does not significantly alter major senescence-associated phenotypes or retinal

function in this model, supporting Bst2 primarily as a selective surface marker rather than a functional driver of senescence.

## Antibody-pluggable and redox-degradable silica nanocarriers were prepared for targeted delivery

Redox-sensitive nanoparticles were designed to degrade efficiently in the intracellular reductive environment, thereby facilitating drug release once internalized[41]. While glutathione (GSH) can be transiently elevated under diverse stress conditions, our data indicates that senescent cells exhibit higher GSH levels compared to normal cells (Supplementary Fig. 3). Thus, the redox-responsive degradation

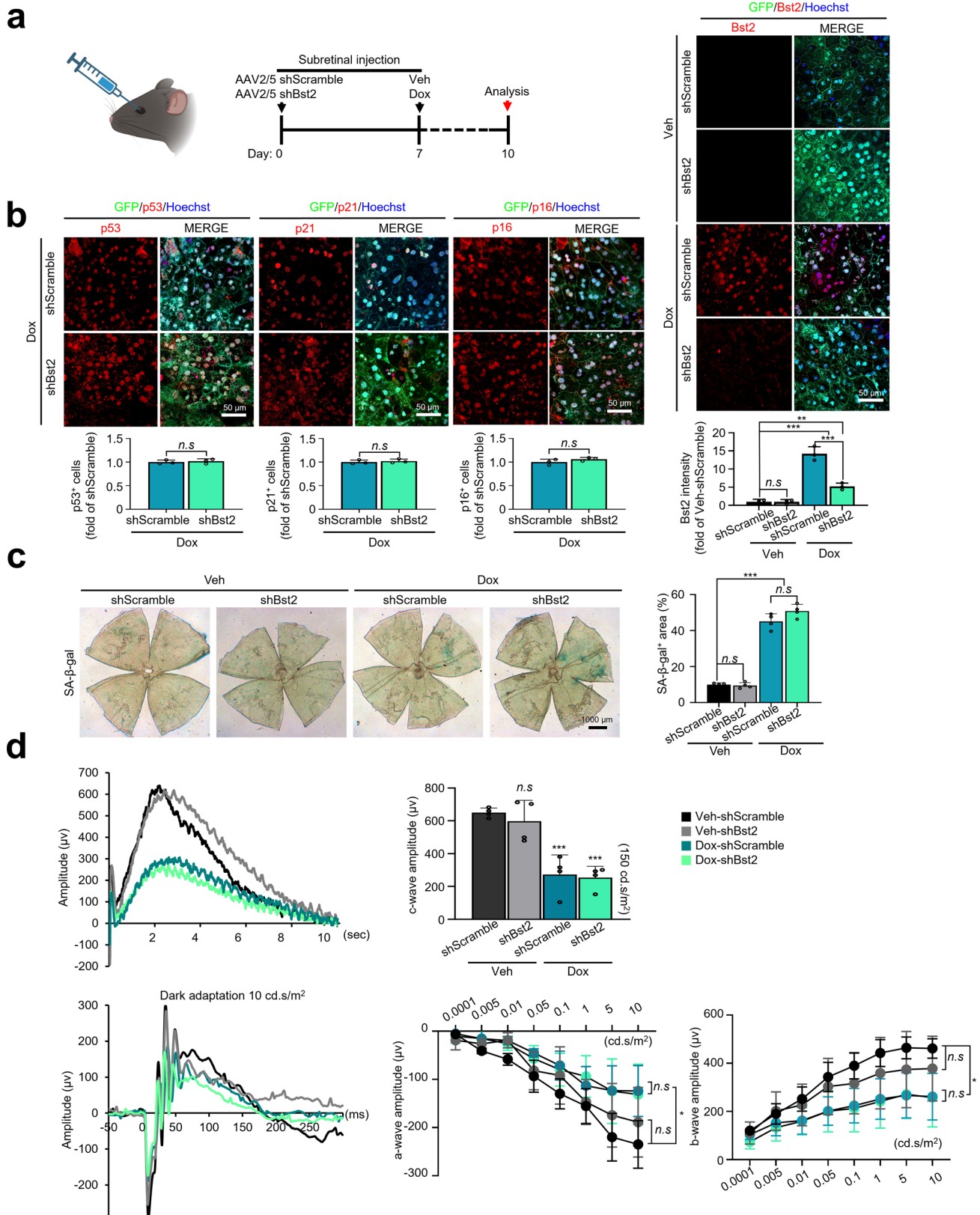

complements the Bst2 antibody-based targeting, ensuring that selective uptake by senescent RPE cells is followed by efficient intracellular drug release. Based on these, we synthesized disulfide-bridged mesoporous organosilica nanoparticles (MONs) as redox-degradable carriers, following a previously established protocol[35] (Supplementary Fig. 4a). To enable GSH-responsive degradation, the MON surface was

subsequently modified with glutathione (GSH-MON). Fourier transform infrared (FT-IR) spectroscopy confirmed successful conjugation, displaying characteristic absorption bands at 1650 cm$^{-1}$ (amide carbonyl stretching from GSH) and 1720 cm$^{-1}$ (ester carbonyl stretching from acrylate) (Supplementary Fig. 4b). To evaluate the degradation behavior of GSH-MONs under senescent-induced cell conditions, GSH-

**Fig. 3 | AAV-mediated Bst2 knockdown in the subretinal Dox-induced RPE senescence model. a** Experimental scheme and representative RPE flat mount images. Mice received subretinal AAV2/5-GFP encoding shScramble or shBst2 at day 0, followed by subretinal vehicle or Dox at day 7 and analysis at day 10. (Created in BioRender. chae, J. (2026) https://BioRender.com/mvoow8l). **b** Representative images show immunofluorescence staining for p53, p21, and p16 (red) in GFP⁺ transduced regions (green) with Hoechst nuclear counterstain (blue), together with Bst2 staining (red) in the same experimental groups. Bar graphs quantify p53⁺, p21⁺, and p16⁺ cells (normalized to Dox+shScramble) and Bst2 fluorescence intensity (normalized to Veh+shScramble), as indicated ($n = 3$ biological replicates). **c** SA-β-gal staining of RPE flat mounts. Quantification of SA-β-gal–positive area showed robust induction by Dox, with no significant difference between shScramble and shBst2 groups (45.1% vs 50.9% relative to shScramble-vehicle control). **d** Electroretinography (ERG) in the same experimental groups. Top, representative c-wave traces and quantification of c-wave amplitude at the indicated stimulus. Bottom, representative dark-adapted traces and quantification of a-wave and b-wave amplitudes across increasing flash intensities (vehicle-shScramble, $n = 4$ eyes; vehicle-shBst2, $n = 4$ eyes; Dox-shScramble, $n = 4$ eyes; Dox-shBst2, $n = 6$ eyes). Scale bars: 50 µm (a), 1000 µm (b). Statistical significance was assessed using one-way ANOVA followed by Tukey's multiple comparisons test for multi-group comparisons and two-tailed unpaired $t$ tests for pairwise comparisons. Data are presented as the mean ± SD. *$P < 0.05$, **$P < 0.01$, and ***$P < 0.001$. Source data are provided as a Source Data file.

MONs and non-responsive silica nanoparticles (GSH-MSNs) were incubated in neutral PBS (10 mM GSH, pH 7.4) at 37 °C with continuous stirring (~120 rpm). Transmission electron microscopy (TEM) analysis revealed substantial structural degradation of GSH-MONs after 72 h, whereas GSH-MSNs remained largely intact, apart from minor aggregation (Supplementary Fig. 4c, d). These results confirm that GSH-MONs undergo selective biodegradation in GSH-rich senescent cells.

Following degradation analysis, GSH-MONs were functionalized with a GST-ABD fusion protein[37] to develop an antibody-pluggable system. GSH-MONs were incubated with GST-ABD at a 1:1 mass ratio in PBS (pH 7.4) under continuous stirring at 4 °C for 1 h, followed by centrifugation to obtain the final protein-conjugated nanocarrier (Z-PON). The surface charge of Z-PON ($-4.11 \pm 1.1$ mV) was comparable to that of free GST-ABD ($-3.18 \pm 1.3$ mV) (Supplementary Fig. 5a). Dynamic light scattering (DLS) measurements further confirmed an increase in hydrodynamic size following modification ($68 \pm 10$ nm for MON, $78 \pm 15$ nm for GSH-MON, and ~$150 \pm 20$ nm for Z-PON) (Supplementary Fig. 5b). TEM imaging showed complete coating of the internal pores of GSH-MON upon GST-ABD conjugation, confirming successful surface functionalization (Supplementary Fig. 5c).

### Bst2-targeted Z-PON enables selective senolysis in vitro

Based on our previously developed antibody plug-and-play system[37], we conjugated Bst2 antibodies to Z-PONs (termed B-Z-PON) via a non-covalent GST-ABD interface, eliminating the need for chemical cross-linking (Fig. 4a). This modular approach enabled straightforward assembly of targeted nanoparticles using Bst2 as a surface anchor enriched in senescence-associated RPE cells.

To assess targeting specificity, a fluorescent dye (DiD) was encapsulated within either antibody-free Z-PONs (DiD@Z-PON) or Bst2-conjugated Z-PONs (DiD@B-Z-PON). Dox-treated ARPE-19 cells were incubated with 30 µL of each formulation (0.5 mg/mL) for 3 h at room temperature. Confocal microscopy revealed markedly enhanced uptake of B-Z-PONs in senescent cells compared to the control, indicating successful targeting (Fig. 4b). Flow cytometry using FITC-labeled nanoparticles confirmed that while DiI@Z-PON uptake was similar in control and senescent cells, DiI@B-Z-PON showed selective accumulation in the senescent cell population (Fig. 4c, d). The enhanced uptake of B-Z-PON in senescent cells is consistent with Bst2-mediated binding and internalization, whereas Z-PON without antibodies shows only baseline, non-specific uptake.

To assess whether Bst2-guided nanoparticles enable selective targeting and senolysis of senescent RPE cells, we loaded the senolytic agent ABT-263 into Z-PON nanocarriers, generating non-targeted (ABT-263@Z-PON) and Bst2-targeted (ABT-263@B-Z-PON) formulations. In control ARPE-19 cells, ABT-263@B-Z-PON exhibited minimal cytotoxicity across matched ABT-263-equivalent concentrations (Fig. 4e). In contrast, in Dox-treated ARPE-19 cultures, which represent a predominantly senescent cell population, ABT-263@B-Z-PON induced a marked reduction in overall viability compared with ABT-263@Z-PON (Fig. 4f). The plateau observed at concentrations above 5 µM likely reflects selective elimination limited by the fraction of Bst2-expressing target cells rather than by drug availability. To further assess nonspecific cytotoxicity relative to nanoparticle delivery, free ABT-263 was additionally tested. In normal ARPE-19 cells, free ABT-263 caused dose-dependent cytotoxicity, with measurable loss of viability at low micromolar concentrations and pronounced toxicity at higher doses, whereas ABT-263@B-Z-PON exhibited minimal toxicity across matched ABT-263-equivalent concentrations (5, 10, and 20 µM; Supplementary Fig. 6). These results indicate that nanoparticle loading substantially mitigates off-target cytotoxicity in normal RPE cells. In Dox-induced ARPE-19 cultures, which represent a mixed population enriched for senescent cells, bulk viability measurements alone are insufficient to distinguish selective senolysis from nonspecific cytotoxicity. Therefore, we further evaluated cell-level senescence and apoptosis markers by immunostaining. p53 signals, elevated in senescent cells, were reduced following ABT-263@Z-PON treatment and more strongly reduced by ABT-263@B-Z-PON (Fig. 4g). Consistently, the proportion of Bst2⁺p53⁺ double-positive cells decreased stepwise after ABT-263@Z-PON and ABT-263@B-Z-PON treatment, supporting preferential elimination of Bst2-high target-positive senescent cells. In parallel, reduction of p21 expression and induction of cleaved caspase-3 (C.C3) within p21⁺ cells further supported apoptosis-associated senolysis (Fig. 4h). In p21⁺ cells, C.C3 positivity increased following ABT-263@Z-PON treatment and was comparatively lower in the ABT-263@B-Z-PON group (Fig. 4h). The reduced C.C3 signal in the ABT-263@B-Z-PON-treated samples likely reflects a more advanced stage of apoptosis, with a substantial fraction of senescent cells already eliminated at the time of analysis. To further substantiate apoptosis-associated senolysis beyond C.C3 staining, we examined canonical Bcl-2 family signaling pathways. Western blot analysis revealed a significant increase in the Bax/Bcl-2 ratio together with a marked reduction in Bcl-xL levels following ABT-263@B-Z-PON treatment (Supplementary Fig. 7a), indicating a shift toward pro-apoptotic signaling in senescent RPE cells. Given that Bcl-xL is a central mediator of apoptotic resistance in senescent cells, its downregulation supports increased susceptibility to apoptosis. These protein-level changes were corroborated at the transcriptional level by qPCR analysis, which demonstrated upregulation of the pro-apoptotic gene Bax and concomitant downregulation of Bst2, following ABT-263@B-Z-PON treatment (Supplementary Fig. 7b). Consistent with these molecular signatures of apoptosis, propidium iodide (PI) staining of RPE flat mounts revealed a significant increase in PI-positive cells after ABT-263@B-Z-PON treatment (Supplementary Fig. 7c), reflecting loss of membrane integrity and increased cell death in senescent RPE. Together, these findings demonstrate that Bst2-targeted nanocarriers can efficiently deliver senolytic agents to selectively eliminate senescence-associated RPE cells while preserving non-senescent counterparts. To further validate its translational relevance, we next investigated the therapeutic potential of this strategy in both chemically induced and naturally aged mouse models of RPE senescence.

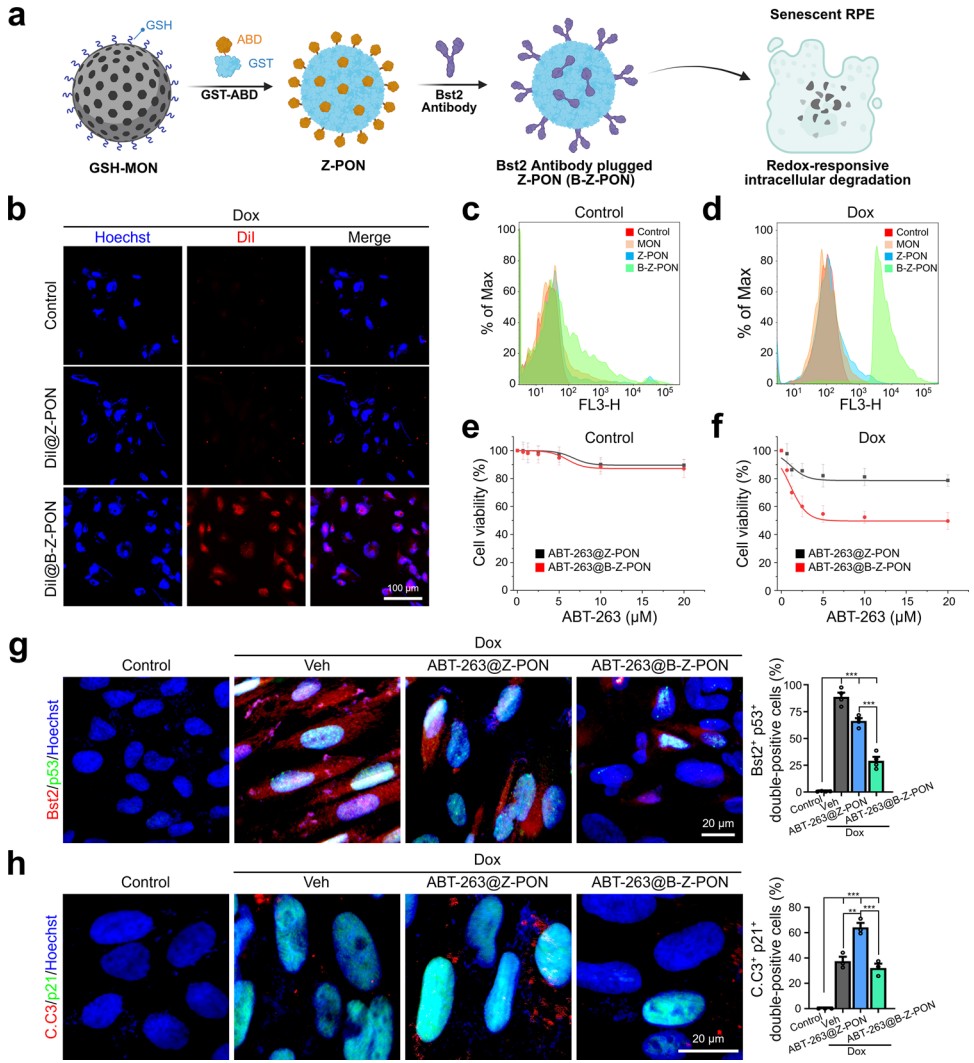

**Fig. 4 | Bst2 antibody-plugged nanoparticle delivery system (B-Z-PON) for selective senolysis in vitro. a** Schematic depiction of the Z-PON platform, consisting of a porous organosilica nanoparticle engineered to allow modular conjugation of targeting antibodies on its surface. Conjugation of an anti-Bst2 antibody generates the B-Z-PON system. The porous framework enables encapsulation of therapeutic agents or fluorescent dyes within the particle interior and supports redox-responsive intracellular degradation in senescent RPE cells. Created in BioRender. Oh, J. Y. (2026) https://BioRender.com/x5khrad. **b** Confocal microscopy images of control and Dox-induced ARPE-19 cells after incubation with DiI-loaded Z-PON (DiI@Z-PON) or DiI-loaded B-Z-PON (DiI@B-Z-PON) for 3 h. Nuclei were counterstained with Hoechst. Flow cytometry analysis of (**c**) control and (**d**) Dox-induced ARPE-19 cells after treatment with FITC-loaded MON, Z-PON, or B-Z-PON. Cell viability (MTT assay) of (**e**) control and (**f**) Dox-induced ARPE-19 cells after

treatment with ABT-263–loaded Z-PON (ABT-263@Z-PON) or ABT-263–loaded B-Z-PON (ABT-263@B-Z-PON) at the indicated concentrations. Immunofluorescence images and quantification of apoptosis-related responses in Dox-induced ARPE-19 cells following treatment with ABT-263@Z-PON or ABT-263@B-Z-PON. Representative images show Bst2 and p53 staining (**g**) or cleaved caspase-3 (C.C3) and p21 staining (**h**), with Hoechst counterstaining (**g**; $n = 4$ biological replicates, **h**; $n = 3$ biological replicates). Quantification shows the percentage of Bst2$^+$p53$^+$ double-positive cells (**g**) or C.C3$^+$p21$^+$ double-positive cells (**h**), calculated as the number of double-positive cells divided by the total number of nuclei per field. Scale bars: 100 μm (**b**), 20 μm (**g**), and 20 μm (**h**). Statistical analyses were performed using one-way ANOVA followed by Tukey's multiple-comparisons test. Data are presented as the mean ± SD. *$P < 0.05$, **$P < 0.01$, and ***$P < 0.001$. Source data are provided as a Source Data file.

## Bst2-pluggable Z-PON alleviates RPE senescence and restores visual function in vivo

To validate Bst2-targeted senolysis in vivo, we administered intravitreal injections of vehicle (PBS), free ABT-263, ABT-263@Z-PON, or ABT-263@B-Z-PON after a stable senescent state had been established (day 7), thereby assessing therapeutic rather than preventive effects. Eyes were analyzed on day 10 (Fig. 5a). As an initial in vivo validation of targeting specificity, nanoparticles were labeled with DiD and delivered intravitreally (DiD@Z-PON). Targeting specificity was conferred by conjugation of a Bst2 antibody to generate DiD-labeled B-Z-PON (DiD@B-Z-PON), with DiD@Z-PON used as a non-targeted control. DiD@B-Z-PON preferentially accumulated in the Dox-induced RPE, whereas DiD@Z-PON showed limited uptake, and little to no signal was

detected in non-senescent regions (Fig. 5b), supporting Bst2-dependent delivery in vivo.

Therapeutically, ABT-263@B-Z-PON markedly reduced RPE senescence markers compared with vehicle and the non-targeted formulation. Immunofluorescence analysis showed a significant decrease in p16-positive RPE cells after ABT-263@B-Z-PON treatment (Fig. 5c). Consistently, immunoblotting demonstrated that ABT-263@B-Z-PON attenuated Dox-induced increases in Bst2 and canonical senescence markers (p53 and p21), bringing their levels toward those observed in controls (Fig. 5d). qPCR analyses further corroborated these findings, showing reduced expression of p53 and p21 transcripts following ABT-263@B-Z-PON treatment (Fig. 5e). Histological analysis (H&E staining) indicated improved retinal integrity, including

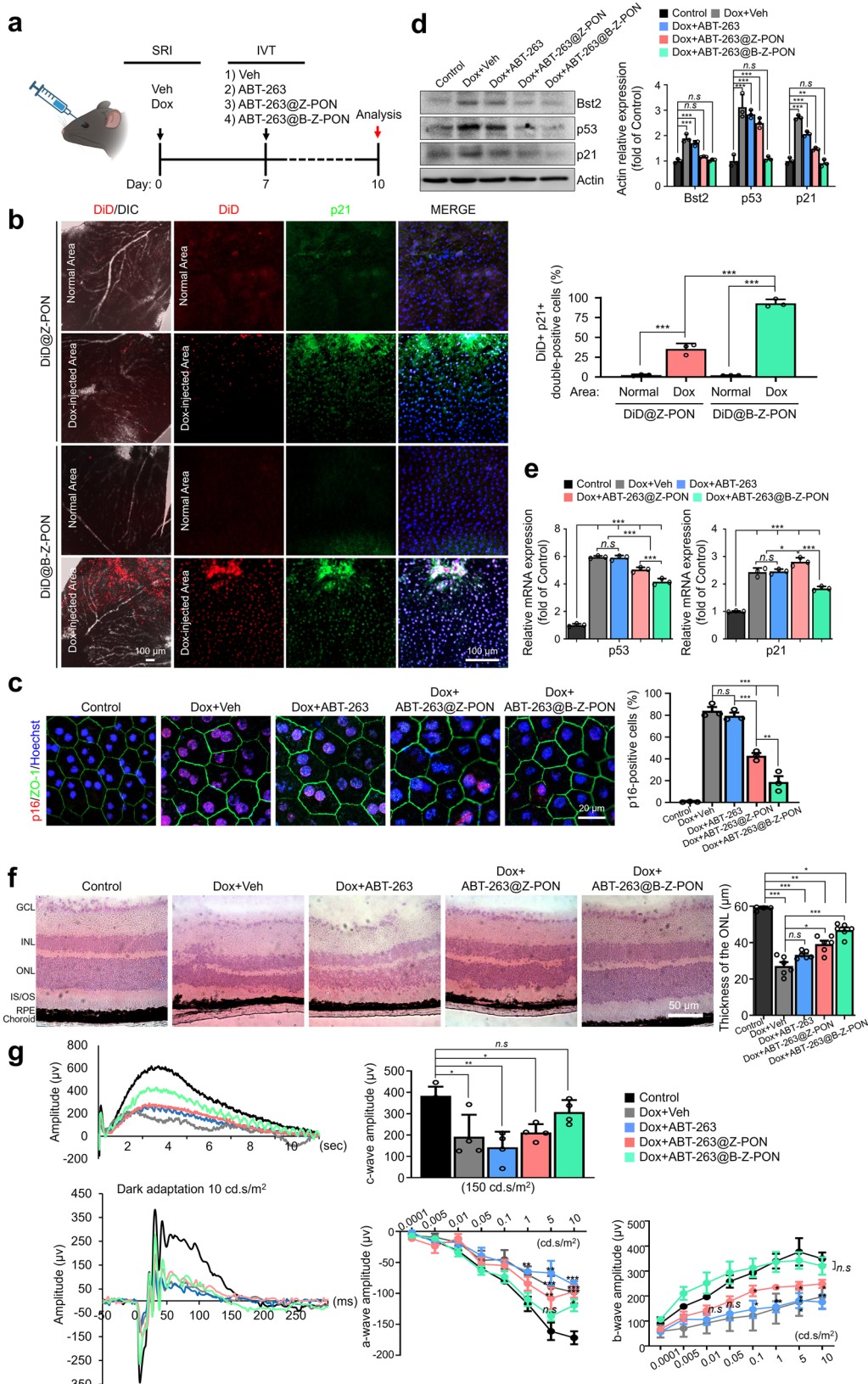

preservation of ONL thickness, in the ABT-263@B-Z-PON group compared with vehicle and ABT-263@Z-PON groups (Fig. 5f). ERG recordings further demonstrated recovery of both c-wave and a-wave amplitudes in the ABT-263@B-Z-PON group, consistent with functional restoration of RPE and photoreceptor activity (Fig. 5g). To further assess the time-dependent senolytic effect, ABT-263@B-Z-PON

was injected on days 1, 3, and 5 post-Dox (Supplementary Fig. 8a). SA-β-gal staining of RPE flat mounts revealed a time-dependent reduction in SA-β-gal positive area, with notable decreases observed on days 3 and 5 following ABT-263@B-Z-PON treatment (Supplementary Fig. 8b). On day 3, p21 mRNA was significantly reduced, while Ki67 expression returned to control levels, suggesting functional rejuvenation

**Fig. 5 | In vivo efficacy of ABT-263@B-Z-PON in the subretinal Dox-induced RPE senescence model. a** Experimental scheme for AAV/particle administration and the subretinal Dox-induced RPE senescence model. After subretinal injection (SRI) of vehicle (PBS) or Dox, mice received intravitreal injections (IVT) of vehicle, ABT-263, ABT-263@Z-PON, or ABT-263@B-Z-PON as indicated, followed by analysis at the defined endpoint. (Fig. 5. Created in BioRender. Oh, J. Y. (2026) https://BioRender.com/w6y8sk3). **b** Representative images of DiD-labeled nanoparticles in RPE flat mounts after intravitreal injection of DiD@Z-PON or DiD@B-Z-PON, showing preferential accumulation in Dox-injected (senescent) areas. p21 immunostaining and DIC are shown for anatomical reference; right, quantification of DiD-positive area in normal versus Dox-injected regions ($n = 3$ biological replicates). **c** Representative confocal images of RPE flat mounts stained for ZO-1 and p16 after treatment with ABT-263@Z-PON or ABT-263@B-Z-PON in Dox-injected eyes. Nuclei were counterstained with Hoechst (blue); right, Quantification shows the percentage of p16-positive cells, calculated relative to the total number of nuclei per field ($n = 3$ biological replicates). **d** Immunoblot analysis of RPE lysates for Bst2 and senescence-associated proteins (p53 and p21), normalized to actin. Dox treatment increased Bst2, p53, and p21 protein levels, whereas ABT-263@B-Z-PON treatment markedly reduced their expression toward baseline levels. Quantification is shown at right ($n = 3$ biological replicates). **e** RT−qPCR analysis of senescence-associated transcripts (p53 and p21) in RPE samples across treatment groups. Dox-induced upregulation of p53 and p21 mRNA was partially reversed following ABT-263@B-Z-PON treatment ($n = 3$ biological replicates). **f** Representative H&E-stained retinal sections across groups; right, quantification of outer nuclear layer (ONL) thickness ($n = 6$ biological replicates). **g** Scotopic electroretinography (ERG) recordings showing representative traces and quantification of c-wave, a-wave, and b-wave amplitudes across experimental groups (all groups, $n = 8$ eyes). Scale bars: 100 μm (**b**), 20 μm (**c**), and 50 μm (**f**). Data are presented as mean ± SD. Sample sizes (eyes) are indicated in the figure or in "Methods". Statistical analyses were performed using one-way ANOVA followed by Tukey's multiple-comparisons test unless otherwise stated. *$P < 0.05$, **$P < 0.01$, ***$P < 0.001$. Source data are provided as a Source Data file.

(Supplementary Fig. 8c). Immunofluorescence revealed increased C.C3 and decreased Bst2 and p16 expression, indicating selective apoptotic clearance of senescent cells between days 0 and 3 (Supplementary Fig. 8d). A vehicle-only time-course after subretinal Dox injection showed that p21 did not decline spontaneously (Supplementary Fig. 9); instead, p21, SA-β-gal activity, and Bst2 increased from day 1 to day 5, supporting progressive senescence and arguing that the p21 decrease after ABT-263@B-Z-PON reflects active senescent-cell clearance. These findings establish Bst2 as a senescence-associated surface marker in RPE degeneration and demonstrate the therapeutic potential of its targeted engagement. ABT-263 delivered via Bst2-pluggable nanoparticles efficiently reversed the transcriptional and protein-level hallmarks of RPE senescence, while sparing non-senescent cells and preserving retinal function.

### ABT-263@B-Z-PON eliminates senescent RPE cells and improves retinal responses in aged mice

Finally, we evaluated the therapeutic efficacy of ABT-263@B-Z-PON in naturally aged mice. Intravitreal injections were administered at three-week intervals, and analyses were performed at 24 months of age after a total of three injections (Fig. 6a). SA-β-gal staining of RPE flat mounts showed a significant reduction in SA-β-gal-positive area after treatment (Fig. 6b). Immunofluorescence analysis of RPE flat mounts and retinal cryosections showed reduced Bst2 and senescence-associated markers (p53 and p21) after ABT-263@B-Z-PON treatment, accompanied by an increase in Ki67-positive RPE cells (Fig. 6c, d). Although mature RPE cells are generally considered post-mitotic, prior studies have reported limited cell-cycle re-entry in response to injury or tissue remodeling[42,43]. Accordingly, the increase in Ki67 signal may reflect a transient cell-cycle–associated response following senescent cell clearance rather than sustained proliferative expansion. Consistent with reduced senescent burden, histological analysis revealed increased ONL thickness on H&E-stained sections, suggesting improved photoreceptor structural preservation (Fig. 6e). ERG recordings further demonstrated an increase in c-wave amplitude, indicating improved RPE function, together with enhanced a- and b-wave amplitudes (Fig. 6f). Collectively, these findings suggest that ABT-263@B-Z-PON reduces the senescent RPE burden in naturally aged mice and is associated with coordinated structural and functional improvements. We summarize a working model in which Bst2 enables phenotype-guided binding and uptake of B-Z-PON into senescent RPE cells, followed by reductive environment-triggered carrier disassembly and ABT-263 release (Fig. 7).

### Discussion

In this study, we identify Bst2 as a senescence-associated surface marker enriched in senescent RPE cells and use it for targeted senolytic delivery. Using Bst2 as a targeting moiety, we developed an antibody–plug-and-play nanoplatform that enables selective delivery of ABT-263 to senescent RPE cells, thereby reducing senescence phenotype and improving functional outcomes in both senescence-induced and naturally aged mouse models. Collectively, these findings support a precision senotherapy strategy that couples senescence-marker discovery with modular intraocular drug delivery to mitigate off-target toxicity and restore RPE homeostasis in age-related retinal degeneration. The clinical need for disease-modifying approaches is particularly pressing in dry AMD and geographic atrophy (GA) where RPE dysfunction and senescence-associated changes are central pathogenic features. Recently approved complement inhibitors (e.g., pegcetacoplan and avacincaptad pegol) can slow GA lesion growth, but functional gains remain limited[44,45], underscoring the need for strategies that target upstream cellular drivers. In this context, selective elimination of senescent RPE cells represents an attractive therapeutic direction, provided that delivery can be confined to the pathogenic cell population within the retina.

ABT-263 is a well-established Bcl-2/Bcl-xL inhibitor with senolytic activity, including in senescent RPE cells as shown in our prior work[22]. However, systemic administration of ABT-263 is limited by dose-limiting toxicities such as thrombocytopenia and neutropenia, which has hindered its broader use for aging-related indications[46,47]. The eye is well suited for local therapy because of its accessibility and compartmentalized anatomy, yet intraocular delivery of free ABT-263 remains challenging due to poor aqueous solubility and limited bioavailability. Consistent with these constraints, intravitreal administration of free ABT-263 produced only modest reductions in senescence markers in our in vivo experiments (Fig. 5). In contrast, encapsulation of ABT-263 into Bst2-targeted, redox-responsive nanoparticles improved effective delivery to senescent RPE cells and enabled intracellular release, resulting in a more pronounced reduction of p53/p21 signals as well as p16, with concomitant functional improvement (Figs. 5, 6). These results highlight the importance of targeted intraocular delivery for realizing the therapeutic potential of senolytics in the retina.

Bst2 is a pleiotropic, interferon-inducible surface protein whose expression is highly context-dependent[31]. A recent study reported Bst2 as a surface marker used to enrich/isolate conjunctival epithelial stem/progenitor-enriched populations in an ocular surface-specific context[48]. In the RPE, we independently validated Bst2 upregulation at the tissue level by immunostaining, demonstrating increased Bst2 signal in aged RPE with co-localization with senescence markers (p53 and p21). In parallel, our scRNA-seq reanalysis showed that Bst2 enrichment within a senescent RPE cluster was accompanied by induction of interferon-stimulated genes and inflammation-associated transcripts, consistent with senescence-associated interferon-like

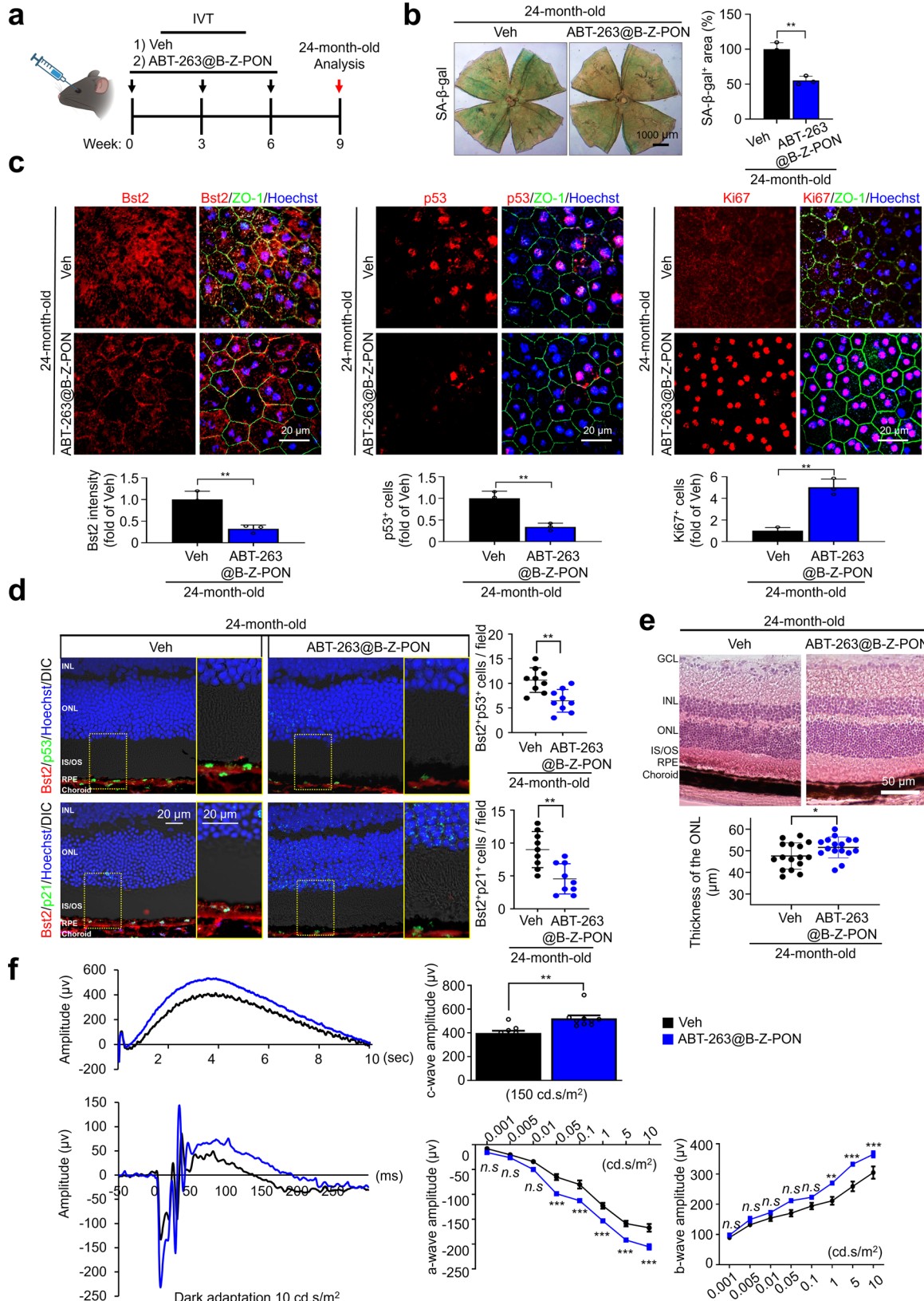

programs[49,50]. Accordingly, in this study, Bst2 is interpreted primarily as a practical senescence-associated targeting marker in the RPE. Consistent with this interpretation, genetic suppression of Bst2 did not alter senescence hallmarks in our models, supporting the view that Bst2 is not required to establish or maintain the senescent phenotype. Moreover, the reduction of Bst2 signal after senolytic treatment is best

explained by preferential elimination of Bst2-high senescent cells, rather than direct pharmacologic repression of Bst2 expression. Supporting translational relevance, multiple independent human RPE transcriptomic datasets showed age-associated increases in BST2 expression in both AMD and non-AMD samples[51,52]. Notably, selective clearance of Bst2-high senescent RPE attenuated senescence-

**Fig. 6 | Improved retinal structure and function following senescent RPE clearance in naturally aged mice. a** Experimental timeline for intravitreal administration in naturally aged mice. Twenty-four-month-old male C57BL/6 J mice received three intravitreal injections of vehicle or ABT-263@B-Z-PON at 3-week intervals, followed by analysis at week 9. (Fig. 6. Created in BioRender. Oh, J. Y. (2026) https://BioRender.com/a575wpw). **b** SA-β-gal staining of RPE flat mounts from 24-month-old mice and quantification of SA-β-gal–positive area ($n = 3$ biological replicates). **c** Representative immunofluorescence images of RPE flat mounts stained for Bst2, p53, or Ki67 together with ZO-1; nuclei were counterstained with Hoechst. Bar graphs show quantification of Bst2 signal intensity and the numbers of p53+ or Ki67+ RPE cells (normalized to vehicle, as indicated, $n = 3$ biological replicates). **d** Representative retinal cryosections showing Bst2 co-localization with

senescence markers (p53, upper; p21, lower) in the RPE layer in vehicle- and ABT-263@B-Z-PON-treated 24-month-old mice. Scatter plots show the numbers of Bst2+p53+ and Bst2+p21+ cells per field within the RPE layer ($n = 9$ biological replicates). **e** Representative H&E-stained retinal sections and quantification of ONL thickness ($n = 8$ eyes). **f** Scotopic electroretinography (ERG) recordings showing representative traces and quantification of c-wave, a-wave, and b-wave amplitudes in vehicle- and ABT-263@B-Z-PON–treated aged mice (both groups, $n = 9$ eyes). Scale bars: 1000 μm (**b**), 20 μm (**c**), 20 μm (**d**), and 50 μm (**e**). Statistical analyses were performed using two-tailed unpaired t tests. Data are presented as the mean ± SD. *$P < 0.05$, **$P < 0.01$ and ***$P < 0.001$. Source data are provided as a Source Data file.

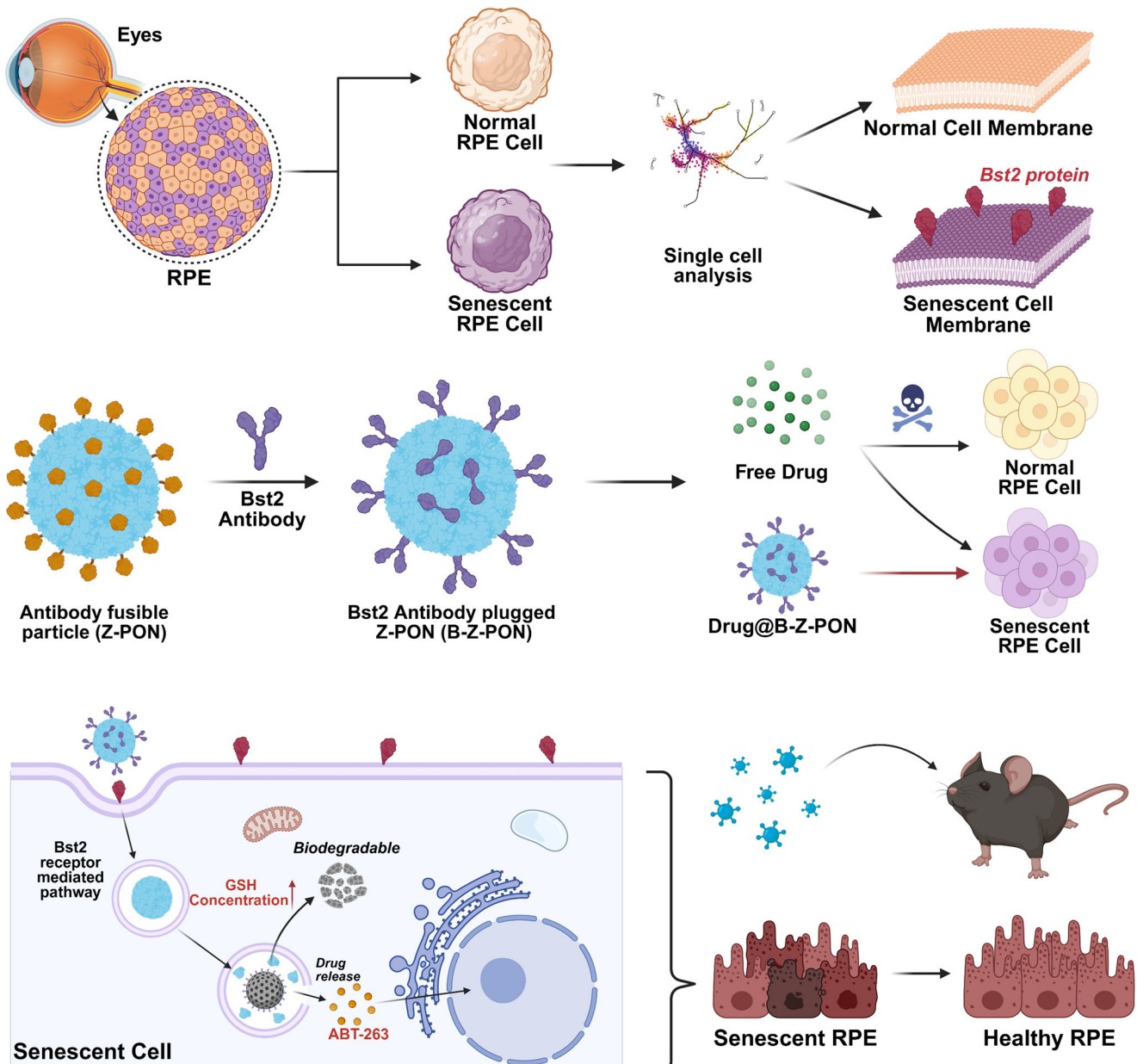

**Fig. 7 | Working model of Bst2-guided nanodelivery to senescent RPE.** Single-cell transcriptomic analyses identify Bst2 as a membrane-enriched marker in senescent RPE, enabling phenotype-guided targeting. A biodegradable mesoporous silica nanoparticle core (MON) is engineered into an antibody-pluggable particle (Z-PON) via immobilized GST–antibody-binding domain (GST–ABD). Bst2 antibodies are then noncovalently assembled on Z-PON through Fc–ABD

interactions to form B-Z-PON, facilitating selective binding and uptake by BST2-high senescent RPE cells. After internalization, the carrier undergoes reductive environment–responsive disassembly, promoting intracellular drug release (e.g., ABT-263) and preferential elimination of senescent RPE cells, supporting restoration of RPE integrity in vivo. Created in BioRender. Oh, J. Y. (2026) https://BioRender.com/fmnx41t.

associated phenotypes and visual function in aged mice. Together, these findings highlight a therapeutic strategy that combines molecular targeting with functional restoration and support the development of safe and effective treatments for retinal aging and dry AMD.

In conclusion, our findings establish Bst2 as a senescence-associated surface marker enriched in senescent RPE cells and show that Bst2-guided intraocular senolytic delivery can reduce senescence-associated burden and restore retinal function in aged mice. This supports precision senotherapy as a feasible strategy for dry AMD and GA.

## Methods

### Ethical statement

All mice were maintained in accordance with guidelines established by the Konkuk University Institutional Animal Care and Use Committee (IACUC) and housed in a specific pathogen-free facility within the Konkuk University Laboratory Animal Research Center. All experimental and animal care procedures were performed according to guidelines approved by the Konkuk University IACUC (KU IACUC; approval No. KU22126).

### Analysis of scRNA-seq

In natural aging mouse (GSE282283), fastq files were mapped to the mouse reference (mm10) dataset. The initial quality control, and the combined gene expression matrix were performed using Cell Ranger pipeline (version 6.0.1, 10X Genomics). The Seurat R package (version 4.1.1) was used to analyze all data after we grouped the count matrix young and aged mice samples. Cells with unique gene counts of less than 200 and with more than 10% of reads originating from mitochondrial genes were filtered. After filtering out cells with zero-sum of hemoglobin gene expression values (*Hbb-bs*, *Hbb-bt*, *Hba-a1*, and *Hba-a2*), a total of 32,285 genes × 27,840 cells matrix were detected. The filtered data were normalized via the NormalizeData function (log-transformation, scale factor = 10,000). We identified 2000 highly variable features using the FindVariableFeatures function (vst method). Following data scaling and principal component analysis (PCA), we clustered the cells using FindClusters (resolution = 0.4) based on the top 25 principal components (PCs). Samples were merged without significant batch effects observed. Visualization was performed using Uniform Manifold Approximation and Projection (UMAP). From the initial 20 clusters, a total of 7908 RPE cells were defined and extracted based on the expression of established markers (*Mitf*, *Rlbp1*, *Rpe65*, *Trpm3*, *Ttr*, and *Tyr*)[51,53]. This isolated population was performed by independent subclustering, including re-identification of 2000 highly variable features and re-computation of PCA. RPE cells using FindClusters (resolution = 0.4) based on the top 11 PCs. Differential gene expression (DGE) analysis between young and aged RPE was conducted using the Wilcoxon rank-sum test via the FindMarkers function (min.pct = 0.1). Significantly differentially expressed genes (DEGs) were defined by a | log2FC | > 0.5 and adjusted $P$value < 0.05. For validation in an independent injury model, we re-analyzed a published scRNA-seq dataset of doxorubicin-injected mouse RPE (GSE183572). The raw data were processed using an analytical pipeline identical to the one described above. Briefly, RPE cells were identified and extracted using the same canonical markers after quality control filters and normalization procedures, resulting in a total of 32,285 genes × 32,392 cells. Subclustering was performed on the isolated RPE population (3423 cells) and DGE analysis identified Dox-induced DEGs (Dox-RPE vs. Control-RPE) using the Wilcoxon rank-sum test, with significance defined by | log2FC | > 0.5 and an adjusted $P$value < 0.05. To define a transcriptomic signature of RPE senescence, we performed an intersection analysis of the upregulated DEGs derived from both models, focusing on candidates with annotated cell-surface localization for downstream validation.

### Experimental animals

Young adult (2 months of age) and aged (21 months of age) Male C57BL/6 J mice were purchased from Orient Bio (Seongnam, Korea). The mice were anesthetized with a mixture of Zoletil (Carros, France) and xylazine (Leverkusen, Germany) (4:1, diluted with normal saline), and pupils were dilated with topical Tropherine eye drops (single use, phenylephrine hydrochloride (5 mg/ml) and tropicamide (5 mg/ml), Hanmi Pharm, Seoul, Korea). An antibiotic ophthalmic ointment (Tarivid, Santen, Osaka, Japan) was applied to all eyes after the procedures.

### Cell culture and senescence induction

The human RPE cell line, ARPE-19 (ATCC, CRL-2302, VA, USA) was cultured in Dulbecco's modified Eagle's medium/nutrient mixture F12 (DMEM/F12; Welgene, LM002-04, Gyeongsangbuk-do, Republic of Korea) containing 10% fetal bovine serum (FBS; Invitrogen, 16000, MA, USA) and 1% penicillin/streptomycin (p/s; Welgene, LS 202–02) at 37 °C in a humidified atmosphere with 5% $CO_2$. Cells with passage numbers 10-20 were used for the experiments. HEK293T (CRL-11268, ATCC) cells were cultured in DMEM supplemented with 10% FBS and were maintained at 37 °C with 5% CO. Both lines were passaged every 2–3 days using trypsin. All data points were measured in triplicate. To induce cellular senescence, ARPE-19 cells were seeded into 24- or 6-well plates at a density of $2.5 \times 10^4$ or $1 \times 10^5$ cells per well. After 24 h, cells were treated with 250 nM Dox (Tocris Bioscience, 2252, Bristol, UK) for 3 days. The medium was then replaced with Dox-free medium, which was changed daily for an additional 4 days before downstream analyses. All data points were measured in triplicate.

**Lentivirus preparation and infection.** To generate lentivirus expressing shScramble (5′-CCTAAGGTTAAGTCGCCCTCG-3′) or BST2 shRNA (5′-GAGGGAGAGATCACTACA TTA-3′), transfer vectors were co-transfected into HEK293T cells along with 2nd generation packaging vectors (psPAX2, pMD2.G) using the standard calcium phosphate transfection method. Following overnight incubation, the cells were washed with PBS, replenished with fresh medium, and maintained at 37 °C with 5% $CO_2$ for an additional 48 h. The viral supernatant was then collected and passed through 0.45-µm pore size filter to remove cellular debris. To concentrate the virus, the supernatant was incubated overnight at 4 °C in the presence of Lenti-X Concentrator. The mixture was subsequently centrifuged at $1500 \times g$ for 45 min, and the resulting viral pellet was resuspended in PBS. The final virus preparation was stored at −80 °C for further use. The generated lentivirus was applied to ARPE-19 cells, and to enhance infection efficiency, polybrene was added at a final concentration of 8 µg/mL. One day post-infection, the culture medium was replaced with fresh medium. Starting from the following day, puromycin was administered at a concentration of 3 µg/mL for two consecutive days to selectively eliminate uninfected cells.

### Quantitative real-time PCR

Total RNA was isolated from ARPE-19 cells and mouse RPE using QIAzol Lysis Reagent (QIAGEN), as previously described[21]. For mouse RPE cells, mice were euthanized and eyes were enucleated immediately. A small incision was made at the junction of the retina and cornea, followed by a complete circumferential cut along the retina–cornea junction to remove the lens and generate an eyecup. The neural retina was carefully removed from the eyecup, leaving the RPE/choroid/sclera complex intact. RPE cells were then selectively lysed using the provided reagent, and total RNA was isolated according to the manufacturer's protocol. Then, SuperiorScript III Master Mix (Enzynomics) was used to reversely transcribe sample RNA to complementary DNAs. Quantitative real-time PCR analysis was performed on the CFX Connect Real-Time PCR Detection System (BIO-RAD) with TOPreal™ SYBR Green qPCR PreMIX (Enzynomics). For the normalization control

GAPDH was used for every sample. See Supplementary Table 1 and Supplementary Table 2 for the list of primers. Organized statistical data were analyzed and plotted using the GraphPad Prism 9.1.0 software (GraphPad Software).

## Western blot analysis

Western blot analysis of ARPE-19 cells and mouse RPE was carried out according to established protocols[21]. Briefly, mouse RPE cells were isolated from eyecups as described above, lysed in RIPA buffer containing protease and phosphatase inhibitors. The tissue lysates were separated on SDS–PAGE gels and detected by immunoblotting with primary antibodies against Bst2 (mouse, 1:1000, Santa Cruz Biotechnology, #sc-390719), p53 (mouse,1:500, Santa Cruz Biotechnology, #sc-126), p21 (mouse, 1:1000, Abcam, #ab109520), p16 (mouse, 1:1000, Abcam, #ab189034), Bcl-2 (rabbit,1:500, Abcam, #ab32124), Bax (mouse,1:200, Santa Cruz Biotechnology, #sc-20067), Bcl-xL (mouse, 1:500, Santa Cruz Biotechnology, #sc-8392), and β-actin (mouse, 1:5000, Santa Cruz Biotechnology, #sc-47778). Based on the types of primary antibodies used, the membranes were incubated with horseradish peroxidase (HRP)-conjugated goat anti-mouse or anti-rabbit secondary antibodies for 2 h. The immunoreactive proteins were detected using an enhanced chemiluminescence substrate (Thermo Fisher, #34580). All experiments were performed at least in triplicate.

## Immunofluorescence of cultured cells, RPE/Choroid flat mounts and cryosectioned retinas

ARPE-19 cells were fixed with 4% PFA for 15 min, whereas RPE flat mounts and cryosectioned retinal tissues were fixed with 4% PFA for 1 h at room temperature. All samples were permeabilized with 0.1% Triton X-100 in PBS for 15 min and incubated with 1% BSA in PBS for 1 h and then overnight at 4 °C with primary antibodies against Bst2 (mouse, 1:500, Santa Cruz Biotechnology, #sc-390719), p53 (mouse, 1:250; Santa Cruz Biotechnology, #sc-126) or p53 (rabbit, 1:500; Cell Signaling Technology, #2527), p21 (mouse, 1:500, Abcam, #ab109520) or p21 (rabbit, 1:500; Cell Signaling Technology, #37543), p16 (mouse, 1:500, Abcam, #ab189034), Ki67 (rabbit, 1:500, Abcam, #ab16667), Cleaved caspase-3 (rabbit, 1:500, Abcam, #ab2302) and ZO-1 (rabbit, 1:1000, Invitrogen, #61-7300, or mouse, 1:1000, Invitrogen, #33-9100). After overnight incubation, the samples were washed with PBS and incubated for 2 h at RT with Alexa Fluor 488- or 555-conjugated goat anti-mouse or anti-rabbit IgG secondary antibodies (1:250; Thermo Fisher Scientific, #11029, #21424, #11034, and #21428). For cryosectioned retinal tissues, lipofuscin autofluorescence was reduced using TrueBlack® Lipofuscin Autofluorescence Quencher (Cell Signaling Technology, #92401) according to the manufacturer's instructions. After incubation with secondary antibodies, the samples were stained with the nuclear dye Hoechst 33342 (1:1000, Thermo Fisher Scientific) in PBS for 15 min at RT. Afterward, the samples were mounted with mounting medium (Polysciences). Immunofluorescence images of ARPE-19 cells, RPE flat mounts, and cryosectioned retinal tissues were acquired using an inverted confocal microscope (Carl Zeiss LSM900, Oberkochen, Germany); for RPE flat mount preparations, images were obtained from the central–equatorial regions. Quantification of immunofluorescence images was performed using Fiji ImageJ. Nuclei were identified using Hoechst staining to determine the total number of cells per field. For single-marker analysis (e.g., p16), the percentage of marker-positive cells was calculated by dividing the number of positive cells by the total number of nuclei. For double-marker analyses (e.g., Bst2 and p53, or C.C3 and p21), cells positive for both markers were counted and expressed as a percentage of total nuclei per field. Threshold-based segmentation was applied to define Bst2 or C.C3 positivity and exclude non-specific signals.

## Senescence-associated β-galactosidase staining

Senescence was assessed using an SA-β-gal staining kit according to the manufacturer's protocol (Abcam, ab102534). Briefly, RPE/choroid tissues were fixed for 1 h in a fixative solution, washed with PBS and incubated overnight with SA-β-gal staining solution at 37 °C incubator. To measure stained SA-β-gal, the dense pigment in RPE was bleached using 10% $H_2O_2$, incubated for 50 min on a 55 °C heat block, and rinsed with PBS. The RPE/choroid tissues were then flat mounted using Aqua Poly/Mount (Polysciences, Inc., 18,606-20, PA, USA) under an optical microscope. Images of the stained RPE/choroid flat mounts were captured using an inverted microscope (Carl Zeiss Axio Scope A1, Gottingen, Germany).

## Injections and eyeball preparation

The subretinal and intravitreal injection procedures were performed as previously described[21]. A total of 1 μL of 100 ng/μL Dox (Tocris Bioscience, 2252, Bristol, UK) was injected into the subretinal space in C57BL/6 J mice. After subretinal injection, 1 μL of vehicle (PBS) or drugs (Free ABT-263 (0.1 mg/mL), Z-PON (0.5 mg/mL, particle), B-Z-PON (0.5 mg/mL, particle), ABT-263@Z-PON (0.1 mg/mL, ABT-263), ABT-263@B-Z-PON (0.1 mg/mL, ABT-263)) was intravitreally injected. For AAV-mediated gene silencing, AAV 2/5 expressing shRNA targeting scramble (5′-CCTAAGGTTAAGTCGCCCTCG-3′) and Bst2 (5′-CTGGA-GAAGAAGGTGTCTCAA-3′) were purchased from VectorBuilder. AAV2/5 particles ($5 \times 10^8$ genome copies/μL, 1 μL) were injected subretinally, following the same injection protocol. After the experiments for analysis, the mice were anesthetized, and their eyeballs were quickly enucleated. The anterior segments were removed from enucleated eyeballs, and the retinas were detached under an optical microscope. The resulting posterior RPE/choroid tissues were washed with cold PBS for RPE analysis.

## Electroretinography (ERG)

Retinal function was assessed in vivo via ERG with a Celeris rodent ERG system (Diagnosys, MA, USA). The mice were adapted to the dark for at least overnight before the experiment. The mice were anesthetized, and pupils were dilated with topical Tropherine ophthalmic solution (Hanmi, Korea), and 2% hypromellose (Samil, Korea) was applied. The mice were kept under dim red illumination during the experiment on a warming plate to maintain their body temperature. For ERG, the electrode was aligned with the center of the pupil in contact with the cornea, and the reference electrode was placed on the contralateral eye. Scotopic ERG responses were stimulated by a single-flash stimulus ranging from 0.001 to 10 cd.s/$m^2$ for a- and b-waves and 150 cd.s/$m^2$ for c-waves. For each flash intensity, a minimum of three responses were recorded and averaged. The amplitudes of a- and b-waves were measured at the maximal negative and positive peaks of the recordings with respect to the baseline before stimulation. The amplitude of the c-wave was measured between the maximal negativity of the c-waves to the maximal peak amplitude of the c-wave.

**Preparation of Z-PON.** Mesoporous organosilica nanoparticles (MONs) were synthesized via a surfactant-templated sol–gel reaction. Cetyltrimethylammonium bromide (CTAB, 1.63 g) and triethanolamine (TEA, 0.29 g) were dissolved in deionized water (106 mL) and heated to 80 °C for 1 h. Tetraethyl orthosilicate (TEOS, 14.3 mL) and bis[3-(triethoxysilyl)propyl] disulfide (BTSPD, 2.4 mL) were then added, and the reaction was continued at 80 °C for 2 h. The resulting particles were collected by centrifugation, washed with deionized water and ethanol, and dried under vacuum. Control mesoporous silica nanoparticles were prepared under identical conditions without BTSPD. Surfactant removal was achieved by acid extraction in methanol (1% HCl) at 60 °C for 24 h, followed by washing and vacuum drying. For surface functionalization, acrylate groups were introduced by reacting MONs

(100 mg, DOX-loaded or unloaded) with 3-(trimethoxysilyl)propyl acrylate (1 mL) in toluene (18 mL) at 60 °C for 24 h. The acrylate-modified MONs were collected by centrifugation and washed with distilled water and ethanol. Glutathione (GSH) conjugation was performed by dispersing the acrylate-modified MONs in N,N-dimethyl-formamide (DMF, 16 mL) containing pyridine (40 μL) and GSH (100 mg dissolved in 2.5 mL of deionized water), followed by stirring at room temperature for 72 h. The resulting GSH-modified MONs (GSH-MONs) were purified by centrifugation, washed with deionized water and ethanol, and dried under vacuum[35]. Subsequently, 1 mg of GSH-MON was dispersed in 5 mL of PBS containing 1 mg of GST-ABD. This mixture was stirred for 1 h at 4 °C. The Z-PONs were then isolated by centrifugation (2300 × g), washed three times with PBS, and finally re-dispersed in 2 mL of PBS.

**Preparation of B-Z-PON.** Bst-2-adsorbed Z-PONs (B-Z-PON) were freshly prepared immediately prior to all experimental procedures. A total of 250 μg of Bst-2 antibody was dispersed in 1 mL of PBS and subsequently added to another 1 mL of PBS containing 0.5 mg of Z-PONs. The resultant mixture was stirred for 15 min at room temperature. Subsequently, the particles were isolated by centrifugation (4600 × g), washed twice with PBS, and re-dispersed in 1 mL of PBS.

**Drug (or dye) loading (%) calculation**
In total, 5 mg of GSH-MON were mixed with 5 mg of ABT-263 (1 mg for DiI, FITC) in DMSO for 48 h. Then centrifuged at 9400 × g for 10 min and collected supernatant. The loading % of the drug was calculated as follows

$$\frac{Drug\,feeding - Supernatant}{Weight\,of\,GSH - MON} \times 100 (\%) \tag{1}$$

**Cellular uptake study.** The cellular uptake of Z-PON and B-Z-PONs in senescent cells was investigated using live confocal microscopy (multiphoton LSM780 confocal microscope with a 40X objective). The cells were seeded at a density of $5 \times 10^4$ cells/well in four-well chambered cover glass (Lab Tek II, Thermo Scientific). After 24 h of incubation, the cells were treated with Z-PON and B-Z-PON samples loaded with DiI dye for 3 h. The cellular uptake of nanoparticles was then observed using confocal fluorescence imaging with the multiphoton LSM780 confocal microscope.

**Flow cytometry analysis.** For quantitative analysis of cellular uptake, cells were seeded at a density of $2 \times 10^5$ cells per well in 6-well plates. After 24 h, cells were treated with FITC-loaded samples and incubated for an additional 3 h. Cells were then collected and analyzed by flow cytometry using a BD FACSVerse flow cytometer (BD Biosciences, USA). Cells were gated based on forward and side scatter to exclude debris and selected the main cell population. FITC uptake was quantified as the mean fluorescence intensity (MFI) of the gated cell population. Unstained cells were used to define background fluorescence. (Supplementary Fig. 10).

**Cytotoxicity study**
ARPE-19 and senescent ARPE-19 cells at 80% confluence were incubated in a 96-well plate (Thermo Fisher) with cell culture medium. The cells were treated with various concentrations of ABT@Z-PON and ABT@B-Z-PON for 24 h. According to the manufacturer's protocol, cell viability was assessed using an MTT assay. The luminescence of the MTT assay was measured using a microplate reader. Results were expressed as percent viability = [(A570 (treated cells)-background)/ (A570 (untreated cells)-background)] × 100.

**Statistical analysis**
All experiments were independently repeated at least three times with similar results unless otherwise stated. All data are presented as the mean ± SD. Statistical significance (P) values were determined using unpaired two-tailed Student's $t$ tests, and multiple datasets were compared by one-way ANOVA followed by Fisher's least significant difference post hoc test or Tukey's multiple comparisons test. All analyses were performed using GraphPad Prism 9.1.0 software (San Diego, CA, USA). Statistical significance was indicated as follows: *$P < 0.05$; **$P < 0.01$; and ***$P < 0.001$.

**Reporting summary**
Further information on research design is available in the Nature Portfolio Reporting Summary linked to this article.

## Data availability

All data supporting the findings of this study are available within the article and its Supplementary Information/Source data file. The scRNA-seq data used in this study are available in the Gene Expression Omnibus (GEO) database under accession codes naturally aged RPE: GSE282283 and doxorubicin-injected RPE: GSE183572. Data are available from the corresponding authors upon request. Source data are provided with this paper.

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

## Acknowledgements

This work was supported by the National Research Foundation of Korea (NRF) grants funded by the Korean Government (Ministry of Science and ICT) (RS-2023-00281553 (J.-H.R.), RS-2025-00553501 (H.C.), RS-2025-16063013 (H.C.), and 2020M3A9D8038192 (J.-H.R.)), by a grant of the Korean ARPA-H Project through the Korea Health Industry Development Institute (KHIDI), funded by the Ministry of Health & Welfare, Republic of Korea (RS-2025-25454860 (H.C.)), and by the InnoCORE program of the Ministry of Science and ICT (GIST InnoCORE KH086030, J.-H.R.). This study contains the results obtained by using the equipment of UNIST Office of Research Facilities and Training (ResFacT).

## Author contributions

H.C. and J.-H.R. conceptualized and designed the study and wrote the manuscript. J.Y.O. designed and synthesized the nanoparticles and contributed to manuscript writing. J.-B.C. performed the in vitro and in vivo experiments and contributed to manuscript writing, with experimental support from C.-W.P. and M.B. H.C., J.-H.R., J.Y.O., J.-B.C., C.-W.P., M.B., C.K., J.J., G.K., and Y.O. contributed to data analysis and interpretation. H.K.L. and S.L. performed the single-cell RNA sequencing data analysis. G.Y., S.K., H.W.O., and D.K. contributed to nanoparticle synthesis and cellular experiments. All authors discussed the results and reviewed and edited the manuscript.

## Competing interests

The authors declare no competing interests.

## Additional information

[1]Department of Chemistry, Ulsan National Institute of Science and Technology (UNIST), Ulsan, Republic of Korea. [2]Department of Ophthalmology, Konkuk University College of Medicine, Seoul, Republic of Korea. [3]Department of Biomedical Engineering, College of Information and Biotechnology, Ulsan National Institute of Science and Technology (UNIST), Ulsan, Republic of Korea. [4]Department of Life Sciences, Pohang University of Science and Technology (POSTECH), Pohang, Republic of Korea. [5]GIST InnoCORE AI-Nano Convergence Initiative for Early Detection of Neurodegenerative Diseases, Gwangju Institute of Science and Technology, Gwangju, Republic of Korea. [6]Fusion Biotechnology, Inc, Ulsan, Republic of Korea. [7]Department of Ophthalmology, Konkuk University Medical Center, Seoul, Republic of Korea. [8]These authors contributed equally: Jun Yong Oh, Jae-Byoung Chae. ✉e-mail: hchung@kuh.ac.kr; jhryu@unist.ac.kr

