## [Transparent Peer Review file · Nature Communications]

Bst2-Targeted Senotherapy Restores Visual Function by Eliminating Senescent Retinal Cells

Corresponding Author: Professor Ja-Hyoung Ryu

Version 0:

Reviewer comments:

Reviewer #1

(Remarks to the Author)

The manuscript by Oh et al is a timely report on a clinically relevant topic, namely the development of second generation senotherapies (in this case, targeted senolytics), which should improve on the efficiency of these drugs by reducing off target effects. The study is a thorough characterization of Bst2 as a marker and target for retinal therapies both in vitro and in vivo, which ends up highlighting the potential of this approach. However, there are a few issues that need to be addressed before its publication.

The main problem with most of the experiments is that there is no attempt to show that the cells to be targeted are senescent. The authors assume that all response to damage is going to result in senescence and they don't even show its establishment. Confirming that the cells are actually senescent is critical for the conclusions of the paper. Thus, the authors need to show that dox-treated RPE cells used in all the experiments are indeed senescent by measuring the phenotype with three independent markers, as is now the consensus. One can be changes of expression in specific markers (however, p53/p21 expression is not sufficient, since these would go up in response to damage regardless of the cell fate induced afterwards), one is usually SA-Bgal and the third one should show that the arrest is indeed irreversible. Moreover, it is unlikely that the doxorubicin treatment induces 100% senescence; therefore, it is important to state what percentage of cells enter the phenotype. Without knowing all this information, it is impossible to assess whether the tools used are indeed senolytic or are they just eliminating cells that have already been damaged previously and are therefore frailer. This is, in fact, what is suggested by figures like 3e, in which practically all cells express p53/p21 and are called senescence (but the response to this kind of stress is very rarely that 100% of the cells go into senescence)

This leads to unsubstantiated claims, like those in figure 3 related to the fact that "Bst2 knockdown does not influence core senescence pathways", based solely on the fact that p53 levels do not change. Bst2 inhibition could be blocking senescent effector pathways below p53 and indeed affecting the phenotype, but this is not explored here.

Similarly, there is little information about the dox-induced retinal mouse model used, presumably because it has been published elsewhere (but this should be referenced, for instance when mentioned at the top of page 9).

Following this thought, in page 9, the authors say that "Senescent cells are hypothesized to exhibit elevated intracellular glutathione (GSH) levels, facilitating the degradation of redox-sensitive nanoparticles". First, there is no reference to this claim (which is, nevertheless, true). Second, all cells will respond with an increase of ROS after being treated with dox, even if they do not enter senescence. Therefore, the specificity of the nanoparticle may not be as high as the authors claim. This needs to be discussed.

After these additional experiments are performed and the issue clarified, the conclusions in each section and the discussion may have to be amended accordingly, since at the moment they do not align with the data shown for the reasons stated above.

OTHER IMPORTANT ISSUES

- Free ABT-263 was tested in control cells (where it had minimal toxicity, despite being used at high doses) in Figure S5 but not in damaged cells. It would be convenient to know how differently dox-treated cells respond to the free drug when

compared to the drug-loaded nanoparticle, in order to establish that the targeted approach is more effective.

- In Figure S6, there is a change in expression in p21 in the ABT-treated cells after being damaged with doxorubicin. However, there is no data shown in the same cells without the senolytic treatment. This is important to rule out that p21 expression decreases on its own after the damage is being resolved.

-The authors use in different places a caspase 3 cleavage as a measurement of induction of apoptosis. However the quantitation of this staining is often done in ways that are not clear. The authors should confirm that apoptosis is induced using a more quantifiable method, such as FACS of AnnexinV/PI-stained cells.

-In Figure 5d, free ABT263 does not seem to be affecting the damaged cells much. Assuming that it has been given at high concentrations (it is not specified in the figure), how do the authors explain this? ABT263 should have a senolytic effect in its free form (which may be enhanced by targeting it with nanoparticles), thus p53/p21 should also go down.

MINOR ISSUES

-The title is confusing (the biomarker and the senotherapy parts are linked by an awkward "and") and does not even mention the target studied. It should feature Bst2 somewhere.

-The introduction is rather short. There are effectively only two actual paragraphs (the rest is an expansion of the abstract, which is not necessary). To set the stage better, it would be convenient to talk first about what are senotherapies (the word is only mentioned once, in the title), the different types (senolytics, senomorphics, senoblockers, senoreversers), the second generation senotherapies (including targeted senolytics), the first attempts with nanoparticles, etc., and end talking about the first clinical trials with senolytics to highlight the relevance of the approach. Without this, the stage for the study is not set properly. In line with this, Figure 1 is a summary of the article and should be better placed at the end of the results or in the discussion.

- In Figure S5, the SA-Bgal staining is difficult to see. The authors should provide better figures and then a quantitation next to them.

-Similarly, the amount of Sa-Bgal positive cells in Figures S6 and 6 should also be quantified using an imaging software, since it is not clear how significant the reduction is.

-In Figure 4 g and h, it is not clear how the data is quantitated. The labels say that the intensity is relative (to what?) and then the units are given as "% of p53 (or p21) positive cells". But one bar is above 100%. How these numbers were achieved should be better explained.

-In Figure 5c, the value of the first bar should not be 1 (like in 5d) if data is expressed as fold changes from this baseline?

Reviewer #2

(Remarks to the Author)

Age-related macular degeneration is a major blinding disease with limited therapeutic options. Senescent RPE cells play an important role in AMD. This study is of significance in that the authors identify a novel senescent RPE cell surface marker, BST2, and develop a BST2-targeted method for delivery of a senolytic drug specifically to senescent RPE cells. This is an interesting and well-designed study. However, it would be appreciated if the authors could address some key points to make their two major conclusions more solid: Bst2 as a novel senescent RPE cell surface marker and BST2-targeted senolytic drug delivery.

A. Establishment of BST2 as a novel senescent RPE cell surface marker. There seems to be conflicting evidence for this.

1. In a recent report (PMC10300367), BST2 has been identified as a specific marker of conjunctival epithelial stem/progenitor cells, which is pro-proliferative. This appears to contradict the upregulation of Bst2 in senescent RPE cells.
2. What is the role of Bst2 in these RPE cells? Although knocking down Bst2 in RPE cells did not affect the levels of senescent RPE cell markers like p16 and p21, would the senolytic drug, which reduces Bst2 as shown in this study, cause undesirable effects in RPE cells?
3. There isn't adequate data showing co-localization of Bst2 and classic senescent cell markers, in vitro or in vivo. For example, in Fig2h, Bst2 and p53 are completely separated from each other in the RPE layer.
4. Bst2 seems to be expressed in all of the RPE cells in Fig2h, in aged mice.
5. Bst2 is also highly expressed in non-RPE cells (IS/OS layer) in aged mice, in Fig2h.
6. In Fig2h, p53, a nuclear protein, does not overlap with the nuclei.
7. Since this study aims at therapeutic development for treating AMD, is there evidence of high expression of Bst2 in senescent RPE cells in human AMD patient samples?

B. Bst2-targeted delivery to senescent RPE cells.

1. ABT263@B-Z-PON is intravitreally injected, but there are retina and other layers of anatomical barriers between vitreous and the RPE layer. How could the bulky nanoparticle with several components including antibody cross these layers and end up being uptaken by RPE cells?

2. Would microglia and other immune cells clear most of the ABT263@B-Z-PON particles?
3. Please provide the evidence that ABT263@B-Z-PON is indeed uptaken by RPE cells of Dox-treated mice and 24-month old mice after intravitreal injection. It is hard to tell from Fig5b whether it is uptaken by senescent RPE cells.
4. What is the evidence that GSH is drastically higher in senescent RPE cells than normal RPE cells?
5. Please explain how ABT263@B-Z-PON can escape from endosomes?
6. In Figs5 and 6, ABT263@B-Z-PON improves ERG a, b, and c waves. Is the retina also morphologically improved? e.g., the outer nuclear layer thickness?
7. Fig6e, in 24-month old mice, RPE cell division should have long ended. Please explain why ABT263@B-Z-PON leads to cell division as shown by Ki67-marked double nuclei in each RpE cell.

Version 1:

Reviewer comments:

Reviewer #1

(Remarks to the Author)

The authors have successfully addressed all my queries.

Reviewer #2

(Remarks to the Author)

It is appreciated that the authors have performed new experiments to obtain essential data, addressed this reviewers' comments in detail, and revised the manuscript carefully. The manuscript has been greatly improved. Kudos to the authors.

Reviewer #1 (Remarks to the Author):

The manuscript by Oh et al is a timely report on a clinically relevant topic, namely the development of second generation senotherapies (in this case, targeted senolytics), which should improve on the efficiency of these drugs by reducing off target effects. The study is a thorough characterization of Bst2 as a marker and target for retinal therapies both in vitro and in vivo, which ends up highlighting the potential of this approach. However, there are a few issues that need to be addressed before its publication.

Response: We thank the reviewers for the positive assessment and constructive comments. We have addressed all points raised and revised the manuscript and supplementary information accordingly, as detailed below. We provide point-by-point responses to each comment.

***Comment 1:** The main problem with most of the experiments is that there is no attempt to show that the cells to be targeted are senescent. The authors assume that all response to damage is going to result in senescence and they don't even show its establishment. Confirming that the cells are actually senescent is critical for the conclusions of the paper. Thus, the authors need to show that dox-treated RPE cells used in all the experiments are indeed senescent by measuring the phenotype with three independent markers, as is now the consensus. One can be changes of expression in specific markers (however, p53/p21 expression is not sufficient, since these would go up in response to damage regardless of the cell fate induced afterwards), one is usually SA-β-gal and the third one should show that the arrest is indeed irreversible. Moreover, it is unlikely that the doxorubicin treatment induces 100% senescence; therefore, it is important to state what percentage of cells enter the phenotype. Without knowing all this information, it is impossible to assess whether the tools used are indeed senolytic or are they just eliminating cells that have already been damaged previously and are therefore frailer. This is, in fact, what is suggested by figures like 3e, in which practically all cells express p53/p21 and are called senescence (but the response to this kind of stress is very rarely that 100% of the cells go into senescence)*

Response: We agree that it is essential to explicitly demonstrate that the doxorubicin (Dox)-treated RPE cells used in this study exhibit a senescence phenotype consistent with established criteria, rather than a transient damage response. We note that we have previously established and characterized a Dox-induced RPE senescence model in our prior work (Refs. #21, 22, and #39). Nevertheless, to directly address the reviewer's concern within the context of the current study, we repeated key validation experiments in ARPE-19 cells and added the results as **new Fig. S2**. ARPE-19 cells were treated with Dox for 3 days, followed by washout and culture in drug-free medium for an additional 4 days (total 7 days). First, SA-β-gal staining confirmed robust senescence induction, with ~84% of cells scoring SA-β-gal-positive at day 7 (**new Fig. S2a**), thereby reporting the fraction of cells that entered the phenotype as requested. Second, to assess durable proliferative arrest after drug removal, EdU labeling and EdU/Ki67 immunostaining were performed during the recovery period; while control cells contained EdU⁺ and Ki67⁺ populations, Dox-treated cells showed no detectable EdU or Ki67 positivity at day 7, consistent with sustained cell-cycle arrest after washout (**new Fig. S2b**). Third, Dox-treated cells showed senescence-associated molecular features beyond p53/p21 induction, including a persistent DNA damage response (significantly increased γH2AX foci per cell versus controls) (**new Fig. S2c**), increased expression of p16 and Bst2 at the protein level (**new Fig. S2d–e**), and upregulation of multiple SASP factors by qRT-PCR (**new Fig. S2f**). Together, these orthogonal parameter (SA-β-gal positivity, sustained post-washout proliferative arrest, and senescence-associated DDR/SASP signatures) confirm that the Dox-treated ARPE-19 cells used in our experiments predominantly adopt a senescent phenotype,

enabling a more rigorous interpretation of senolytic activity. We have incorporated these data into the **Fig.S2** and added a brief description in the main text where the Dox-induced senescence model in vitro is introduced (**lines 159-163 in page 7**).

Comment 2: This leads to unsubstantiated claims, like those in figure 3 related to the fact that “Bst2 knockdown does not influence core senescence pathways”, based solely on the fact that p53 levels do not change. Bst2 inhibition could be blocking senescent effector pathways below p53 and indeed affecting the phenotype, but this is not explored here.

Response: We appreciate the reviewer’s important point. We agree that unchanged p53 levels alone are insufficient to conclude that “core senescence pathways” are unaffected. To address this concern, we expanded our analyses to include multiple orthogonal senescence readouts beyond p53 and performed additional experiments in our Dox models, both in vitro and in vivo. These new data show that Bst2 knockdown does not significantly change major senescence-associated phenotypic parameters in these settings, supporting Bst2 primarily as a senescence-associated surface marker rather than a functional driver in our models. We added the details and corresponding data in the revised version. Specifically, we assessed SA-β-gal activity, canonical senescence marker expression (p53/p21/p16), and persistent DNA-damage signaling (γH2AX foci), and evaluated functional outcomes in vivo.

In Dox-treated ARPE-19 cells, we compared shScramble and shBst2 conditions using multiple senescence parameters beyond p53. SA-β-gal positivity was not significantly different between shScramble and shBst2 senescent cells (**new Fig. 2b,c**). Consistently, immunoblotting showed no significant differences in canonical senescence markers (p53, p21, and p16) between groups (**new Fig. 2d**). In addition, immunofluorescence analyses of p53/p21/p16 and persistent DDR (γH2AX foci) likewise revealed no significant differences (γH2AX foci: shBst2 23.4 ± 7.6 vs shScramble 22.1 ± 9.1 foci/cell; **new Fig. 2e**). Collectively, these data indicate that Bst2 knockdown does not measurably change the establishment or maintenance of the senescent phenotype in this in vitro setting.

We further validated this conclusion in vivo using the Dox-induced mouse RPE senescence model. As expected, Dox administration increased senescence parameters, including p16 immunoreactivity and SA-β-gal activity, compared with vehicle controls. However, within the Dox-treated groups, there were no significant differences between shScramble and shBst2, not only in p53-associated markers (p53 and p21) but also in p16 (**new Fig. 3a,b**). Together, these data indicate that Bst2 knockdown does not measurably alter major senescence marker expression beyond p53 alone in this model. To assess functional consequences, we performed ERG analyses. Dox treatment reduced the c-wave (reflecting RPE function) as well as a- and b-waves, yet Bst2 knockdown neither improved nor worsened these responses compared with shScramble within the Dox condition (**new Fig. 3d**). Based on these expanded in vitro and in vivo datasets, we now state more precisely that Bst2 knockdown does not significantly alter the establishment or maintenance of major senescence-associated phenotypic parameters-including SA-β-gal activity, p53/p21/p16 expression, and γH2AX foci-nor retinal functional outcomes in the Dox models. These data are included in **revised Figures 2 and 3**, and the corresponding text has been updated accordingly (**lines 170-189 in page 7**).

Comment 3: Similarly, there is little information about the dox-induced retinal mouse model used, presumably because it has been published elsewhere (but this should be referenced, for instance when mentioned at the top of page 9).

Response: We appreciate the reviewer's comment and agree that the Dox-induced mouse RPE senescence model should be more clearly described and appropriately referenced at first mention. This model was previously established and comprehensively validated in our prior work (Ref. #21), and it has also been used as an *in vivo* platform for senolytic evaluation in our subsequent study (Ref. #22,38,39). In the revised manuscript, we have now added these references at the first mention of the model (**lines 135-137, page 5**) and incorporated the experimental scheme and key validation/parameters in the corresponding Results and figure panels. Briefly, localized RPE senescence is induced by subretinal injection of Dox (100 ng/ μ L), and senescence is assessed in RPE flat mounts within the established time window (minimal at day 3, peak at day 7, declining by day 14, as reported in Ref. #21). The model shows RPE-specific SA- β -gal induction, increased expression of senescence markers (e.g., p53/p21/p16) in the RPE tissue, senescence-associated morphological changes, and minimal apoptosis by TUNEL, supporting senescence rather than nonspecific cytotoxicity (Ref. #21). For the reviewer's convenience, we have attached the key validation data from Ref. #21 (previously Supplementary Fig. 4) below.

Our published data (Ref. #21, Chae et al., GeroScience, 2021, Supplemental Fig. 4. Dox induces the senescence of RPE cells *in vivo*.)

Supplemental Fig. 4. Dox induces the senescence of RPE cells *in vivo*.

Comment 4: Following this thought, in page 9, the authors say that “Senescent cells are hypothesized to exhibit elevated intracellular glutathione (GSH) levels, facilitating the degradation of redox-sensitive nanoparticles”. First, there is no reference to this claim (which is, nevertheless, true). Second, all cells will respond with an increase of ROS after being treated with dox, even if they do not enter senescence. Therefore, the specificity of the nanoparticle may not be as high as the authors claim. This needs to be discussed.

Response: We thank the reviewer for this important point. We agree that elevated intracellular glutathione (GSH) is not specific to senescence and can be induced by diverse stress responses, including Dox exposure. We have therefore revised the manuscript to avoid implying that redox responsiveness alone confers senescence specificity. Instead, the redox-responsive component is intended to facilitate intracellular drug release under reductive conditions once nanoparticles are internalized, whereas cell-type/phenotype selectivity is primarily provided by Bst2 antibody-mediated targeting. We also added an appropriate reference supporting that intracellular GSH/redox homeostasis can be altered in certain senescence models (Ref. #41), and we briefly discuss the limitation that stressed but non-senescent cells may also show increased GSH. We revised the corresponding text in the Results section to reflect this clarification (**lines 193-196, page 9**).

After these additional experiments are performed and the issue clarified, the conclusions in each section and the discussion may have to be amended accordingly, since at the moment they do not align with the data shown for the reasons stated above.

Response: We thank the reviewer for this important and constructive comment. Following the reviewer’s guidance, we performed all experiments requested to clarify the points raised, and we subsequently re-evaluated our interpretation of the results. We have revised the conclusions in the relevant sections and updated the Discussion to ensure that they accurately reflect the data, including refining statements that were previously overstated or not sufficiently supported. We believe these revisions have improved the consistency between the experimental evidence and our conclusions. Thank you.

OTHER IMPORTANT ISSUES

Comment 5: Free ABT-263 was tested in control cells (where it had minimal toxicity, despite being used at high doses) in Figure S5 but not in damaged cells. It would be convenient to know how differently dox-treated cells respond to the free drug when compared to the drug-loaded nanoparticle, in order to establish that the targeted approach is more effective.

Response: We thank the reviewer for this important suggestion and agree that a direct comparison between free ABT-263 and nanoparticle-loaded ABT-263 (ABT-263@B-Z-PON) in Dox-treated cells might be informative for evaluating the relative effects of the two formulations.

First, we would like to clarify and correct a labeling error in the previous version: in Fig. S5a (**now Fig. S6a**), the treatment conditions were mislabeled. We have corrected the labels to accurately indicate free ABT-263 versus ABT-263-loaded B-Z-PON (ABT-263@B-Z-PON), and we apologize for the confusion. To directly address the reviewer’s request, we repeated the cytotoxicity (MTT) experiments with increased rigor, including independent biological replicates (n = 3–5 per condition), and performed quantitative analyses.

In normal ARPE-19 cells, free ABT-263 showed clear dose-dependent cytotoxicity, with measurable toxicity already at low micromolar concentrations, whereas ABT-263@B-Z-PON exhibited significantly reduced toxicity at the same nominal ABT-263-equivalent concentrations

(**updated Fig. S6a**). These results confirm that nanoparticle loading markedly attenuates nonspecific cytotoxicity in non-senescent RPE cells. In addition, and specifically in response to the reviewer's concern, we performed a head-to-head comparison of free ABT-263 and ABT-263@B-Z-PON in Dox-induced ARPE-19 cultures (1, 5, 10, and 20 μM ABT-263-equivalent concentrations; $n = 3$; new data shown in this response letter below as **Figure R1**). These cultures represent a senescence-enriched but heterogeneous cell population. As expected, both free ABT-263 and ABT-263@B-Z-PON induced substantial reductions in overall cell viability, reflecting the intrinsic senolytic activity of ABT-263. Notably, free ABT-263 consistently produced a more profound loss of viability across doses, consistent with its well-documented non-selective cytotoxic effects. Importantly, however, we emphasize that bulk viability assays such as MTT cannot distinguish selective senolysis from generalized cytotoxicity, particularly in senescence-enriched cultures. Free ABT-263 is known to induce apoptosis in both senescent and non-senescent cells, making bulk viability readouts insufficient to establish selectivity. For this reason, we do not use MTT data from Dox-treated cultures as the primary evidence for selective senolysis, and these data are provided for completeness in the response letter rather than the main figures. Instead, the added value of the targeted nanoparticle formulation is supported by multiple complementary lines of evidence. At the cellular level, ABT-263@B-Z-PON preferentially reduced Bst2-high and p53/p21-high cell populations, as demonstrated by immunofluorescence analyses (**Fig. 4g, h**), indicating selective elimination of senescent cells rather than indiscriminate cytotoxicity. Importantly, our *in vivo* experiments provide critical validation in the Dox-induced mouse RPE model: when free ABT-263 and ABT-263@B-Z-PON were administered intravitreally at matched ABT-263-equivalent doses (50 μM), ABT-263@B-Z-PON achieved a significantly greater reduction of senescence markers in the Dox-induced RPE without detectable nonspecific toxicity, whereas free ABT-263 did not show comparable efficacy (**Fig. 5**). Collectively, these expanded *in vitro* and *in vivo* analyses demonstrate that although free ABT-263 can reduce cell viability more strongly in senescence-enriched cultures, ABT-263@B-Z-PON provides superior therapeutic selectivity and a more favorable efficacy. We have revised the Results and Discussion sections accordingly to reflect these points (**lines 248-257, page 12; line 258, page 13; lines 376-384, page 24**).

Figure R1. Bulk viability analysis of free ABT-263 and ABT-263@B-Z-PON in Dox-induced ARPE-19 cells.

Comment 6: In Figure S6, there is a change in expression in p21 in the ABT-treated cells after being damaged with doxorubicin. However, there is no data shown in the same cells without the senolytic treatment. This is important to rule out that p21 expression decreases on its own after the damage is being resolved.

Response: We thank the reviewer for raising this important point. We agree that it is necessary to rule out the possibility that p21 decreases spontaneously as Dox-induced damage resolves. To address this, we performed a time-course analysis in the RPE of Dox-injected mice without senolytic treatment (vehicle control) and added these new data as **new Fig. S9**, entitled “Time course of senescence-marker induction after subretinal Dox injection,” following Fig. S8 (previously Fig. S6 mentioned in this comment). In this vehicle-only group, p21 did not decline over time; instead, p21 expression remained elevated and increased from day 1 to day 5 (qPCR and immunofluorescence). Consistently, SA- β -gal activity increased over the same period, and Bst2 expression also showed a time-dependent increase, supporting progressive establishment of the senescent phenotype rather than spontaneous recovery. Therefore, the reduction of p21 observed in the ABT-263@B-Z-PON-treated group is unlikely to be explained by natural resolution of Dox-induced stress and is more consistent with active removal of senescent RPE cells. We added this briefly in the Results section (**lines 319-321, page 16; line 322, page 17**).

Comment 7: The authors use in different places a caspase 3 cleavage as a measurement of induction of apoptosis. However the quantitation of this staining is often done in ways that are not clear. The authors should confirm that apoptosis is induced using a more quantifiable method, such as FACS of AnnexinV/PI-stained cells.

Response: We appreciate the reviewer’s comment and agree that apoptosis should be supported by clear and quantifiable measures. In our study, cleaved caspase-3 staining was used primarily to localize apoptotic events within the RPE layer rather than as the sole quantitative measure, and we have clarified the quantification strategy in Fig. 4h and Fig. S8d and revised the corresponding figure legends. To provide additional quantifiable support for apoptosis-mediated senolysis, we added new analyses of canonical apoptotic signaling and membrane integrity in the Dox-induced senescent RPE model (**new Fig. S7**). Specifically, ABT-263@B-Z-PON treatment shifted Bcl-2 family signaling toward apoptosis (increased Bax/Bcl-2 balance and reduced Bcl-xL protein levels in RPE lysates), with consistent changes at the mRNA level (**Fig. S7a,b**). In addition, we performed propidium iodide (PI) staining on RPE flat mounts and quantified PI-positive RPE cells as a complementary in vivo endpoint indicative of loss of membrane integrity in dying/dead cells (**Fig. S7c**). While Annexin V/PI flow cytometry is highly informative for in vitro apoptosis quantification, it is technically challenging to apply to in vivo RPE because the RPE is a thin monolayer with limited total cell yield, and tissue dissociation can reduce viability and introduce processing-related artifacts. Therefore, we relied on complementary quantitative endpoints—viability reduction in vitro and reduced senescence burden in vivo (SA- β -gal area and senescence markers)—together with the newly added Bcl-2 family and PI analyses to substantiate apoptosis-associated senolysis. We revised the Results section accordingly (**lines 268-279, page 13**).

Comment 8: In Figure 5d, free ABT263 does not seem to be affecting the damaged cells much. Assuming that it has been given at high concentrations (it is not specified in the figure), how do the authors explain this? ABT263 should have a senolytic effect in its free form (which may be enhanced by targeting it with nanoparticles), thus p53/p21 should also go down.

Response: We thank the reviewer for raising this important point. We agree that ABT-263 is a well-recognized senolytic agent; however, its efficacy in free form is constrained by intrinsic pharmacological limitations. ABT-263 is highly hydrophobic with poor aqueous solubility, which restricts its effective concentration in the intraocular environment. In addition, as reported in previous studies including ours (Ref. #21), free ABT-263 suffers from limited bioavailability and dose-limiting systemic toxicities, all of which reduce its functional impact in vivo. These factors likely account for the only modest changes in p53/p21 observed in the RPE from Dox-treated mice after free ABT-263 administration in vivo. By contrast, our nanoparticle-based system directly addresses these shortcomings. Encapsulation of ABT-263 within Bst2-targeted, redox-responsive nanocarriers enhances drug solubility, facilitates specific uptake into senescent RPE cells, and promotes efficient intracellular release. Consequently, ABT-263@B-Z-PON treatment produced a marked decrease in senescence markers and restoration of retinal function, underscoring the critical importance of nanomedicine-enabled targeting for therapeutic efficacy (**Fig. 5**). We have clarified this rationale in the revised Discussion (**lines 376-384, page 24**).

MINOR ISSUES

Comment 9: The title is confusing (the biomarker and the senotherapy parts are linked by an awkward “and”) and does not even mention the target studied. It should feature Bst2 somewhere.

Response: We thank the reviewer for this valuable suggestion. We agree that the original title was unclear and did not highlight the biomarker studied. We have revised the title to explicitly feature Bst2 and to more clearly connect biomarker-based targeting with senotherapy:

Revised Title: Bst2-Targeted Senotherapy Eliminates Senescent RPE Cells and Improves Visual Function

Comment 10: The introduction is rather short. There are effectively only two actual paragraphs (the rest is an expansion of the abstract, which is not necessary). To set the stage better, it would be convenient to talk first about what are senotherapies (the word is only mentioned once, in the title), the different types (senolytics, senomorphics, senoblockers, senoreversers), the second generation senotherapies (including targeted senolytics), the first attempts with nanoparticles, etc., and end talking about the first clinical trials with senolytics to highlight the relevance of the approach. Without this, the stage for the study is not set properly. In line with this, Figure 1 is a summary of the article and should be better placed at the end of the results or in the discussion.

Response: We thank the reviewer for this constructive comment. In response, we substantially expanded and reorganized the Introduction to better set the therapeutic landscape of cellular senescence and senotherapies. The revised Introduction now briefly defines senotherapies and focuses on the modalities most relevant to this study (senolytics and senomorphics), while introducing the rationale for second-generation/precision senotherapy, including targeted senolytics and nanomedicine-enabled delivery to improve selectivity and safety. We also added a concise paragraph highlighting early clinical translation of senolytics, including an intermittent dasatinib plus quercetin regimen reported to reduce senescent cell burden in patients with diabetic kidney disease (Ref. #29) and intravitreal administration of a BCL-xL-targeting senolytic evaluated in diabetic macular edema (Ref. #30). These clinical studies underscore the translational relevance of senolytic strategies while reinforcing the need for improved tissue specificity and local delivery. As suggested by the reviewer, we moved Figure 1 to the end of the Results section as a working model (**now Fig. 7, lines 347-350 of page 20**). For the reviewer’s

convenience, **all revisions in the Introduction are highlighted in blue in the revised manuscript.**

Comment 11: In Figure S5, the SA-Bgal staining is difficult to see. The authors should provide better figures and then a quantitation next to them.

Response: We thank the reviewer for this helpful comment. We have replaced **Fig. S6** with higher-contrast representative images and added quantitative analysis alongside the images. As this experiment was designed to test whether ABT-263 induces senescence in normal ARPE-19 cells, SA- β -gal staining remained minimal/absent, consistent with a negative result rather than a technical issue. We revised the **Fig. S6 legend** accordingly.

Comment 12: the amount of Sa-Bgal positive cells in Figures S6 and 6 should also be quantified using an imaging software, since it is not clear how significant the reduction is.

Response: We appreciate the reviewer's comment. In the revised manuscript, we added quantitative analyses for SA- β -gal staining for all figures that include SA- β -gal images (including **Figs. 3, 6 and Fig. S8, S9** in revised manuscript), and we also added quantification for the relevant immunofluorescence images. For SA- β -gal, we quantified the SA- β -gal-positive area using ImageJ with a consistent, threshold-based workflow applied uniformly across all groups, and we now present the results with statistical analysis. This ImageJ-based quantification follows the workflow we previously established and reported in our prior work (Ref. #21), which provides a step-by-step description and representative examples of the ImageJ processing pipeline (see Supplemental Fig. 3 in Ref. #21).

Supplemental Fig. 3. Quantification of SA- β -gal staining. All image processing steps were performed using ImageJ software (v. 1.50i). First, we resized all photographs to the same size. Then, the 'Split Channels' function was used to separate each photograph in RGB color into three photographs composed of the red, green or blue channel. We used the red channel photograph to increase the contrast between the SA- β -gal-positive area and the background. Then, the darker area originating from SA- β -gal was binarized with the threshold function of ImageJ. The boundary between the stained and unstained areas was manually demarcated. Then, the pixels inside the stained area were counted with the 'Measure' function and converted into the area as mm².

Comment 13: In Figure 4 g and h, it is not clear how the data is quantitated. The labels say that the intensity is relative (to what?) and then the units are given as “% of p53 (or p21) positive cells”. But one bar is above 100%. How these numbers were achieved should be better explained.

Response: We thank the reviewer for pointing out the lack of clarity in the quantification and labeling of Fig. 4g and h. We apologize for the confusion in the previous version. Nuclei were identified using Hoechst staining to define the total number of cells per field. Bst2 (or cleaved caspase-3) positivity was determined in the red channel using threshold-based segmentation to exclude non-specific signals, while p53 or p21 positivity was identified by counting marker-positive nuclei in the green channel. Cells positive for both markers were counted as double-positive cells. The y-axis therefore represents the percentage of double-positive cells relative to the total number of cells in each group. We have revised the y-axis labels and figure legends accordingly to clearly reflect this cell-based quantification method, and we updated the Methods section to describe the analysis (lines 542-548, page 29).

Comment 14: In Figure 5c, the value of the first bar should not be 1 (like in 5d) if data is expressed as fold changes from this baseline?

Response: We thank the reviewer for raising this point and apologize for the confusion caused by the previous labeling. The data shown in Fig. 5c are not expressed as fold changes, but represent a cell-based percentage quantification. Specifically, p16 positivity was quantified by counting the number of p16-positive cells and dividing it by the total number of nuclei (Hoechst-positive cells) per field for each group. Accordingly, the y-axis represents the percentage of p16-positive cells,

and the first bar is not constrained to equal 1. We have revised the y-axis label and figure legend to clearly reflect this definition and updated the Methods section accordingly (**lines 542-548, page 29**).

Reviewer #2 (Remarks to the Author):

Age-related macular degeneration is a major blinding disease with limited therapeutic options. Senescent RPE cells play an important role in AMD. This study is of significance in that the authors identify a novel senescent RPE cell surface marker, BST2, and develop a BST2-targeted method for delivery of a senolytic drug specifically to senescent RPE cells. This is an interesting and well-designed study. However, it would be appreciated if the authors could address some key points to make their two major conclusions more solid: Bst2 as a novel senescent RPE cell surface marker and BST2-targeted senolytic drug delivery.

Response: We thank the reviewer for the thoughtful and encouraging assessment of our study and its significance to AMD. We appreciate the opportunity to further strengthen our two central conclusions—(i) Bst2 as a senescence-associated RPE surface marker and (ii) Bst2-targeted senolytic drug delivery. In the revised manuscript, we have incorporated additional validation and quantitative analyses and clarified key methodological details, as outlined below in our point-by-point responses.

Comment Section A. Establishment of BST2 as a novel senescent RPE cell surface marker. There seems to be conflicting evidence for this.

***Comment 1:** In a recent report (PMC10300367), BST2 has been identified as a specific marker of conjunctival epithelial stem/progenitor cells, which is pro-proliferative. This appears to contradict the upregulation of Bst2 in senescent RPE cells.*

Response: We thank the reviewer for raising this important point. We agree that BST2 has been reported as a surface marker used to enrich/isolate conjunctival epithelial stem/progenitor-enriched populations in a tissue-specific context (PMC10300367) and we have added this study as Ref. #48 in the revised manuscript. However, this does not contradict our findings in the RPE. Bst2 is a pleiotropic, interferon-inducible surface protein whose expression is highly context-dependent and can be induced by innate immune activation, inflammation, and cellular stress responses. In line with this biology, senescent cells can acquire an antiviral-like/type I interferon state downstream of persistent stress and DNA damage responses, leading to upregulation of interferon-stimulated genes (ISGs) that may include Bst2. Consistent with this, in our scRNA-seq reanalysis, Bst2 enrichment within a senescent RPE cluster was accompanied by induction of ISGs and inflammation-associated transcripts, consistent with senescence-associated interferon-like programs, supporting that Bst2 upregulation reflects a stress/innate immune-associated senescence program in the RPE context rather than a universal stemness signature. Moreover, Bst2 upregulation in our in vitro and in vivo models coincided with established senescence hallmarks; in response to the reviewer's related comment (#3), we performed additional validation experiments and **updated Figs. 2 and 3** accordingly. Together, these findings further support Bst2 as a practical senescence-associated surface marker for targeted delivery in the RPE. We incorporated this point into the revised Discussion and added refs. #48-50. (**lines 385-388, page 24; lines 389-399, page 25**).

Comment 2: What is the role of Bst2 in these RPE cells? Although knocking down Bst2 in RPE cells did not affect the levels of senescent RPE cell markers like p16 and p21, would the senolytic drug, which reduces Bst2 as shown in this study, cause undesirable effects in RPE cells?

Response: We sincerely thank the reviewer for this thoughtful question. Our data suggest that Bst2 is unlikely to be a causal regulator of RPE senescence; rather, it is best interpreted as a surface marker that becomes upregulated under senescence-associated conditions. Importantly, the apparent “reduction” of Bst2 signal after senolytic treatment is best explained by preferential elimination of Bst2-high senescent RPE cells, rather than a direct adverse effect of the drug on normal RPE homeostasis. To directly address whether Bst2 suppression itself alters the senescence program or could be deleterious to RPE cells, we additionally performed Bst2 knockdown experiments during the revision, both in vitro and in vivo. In Dox-treated ARPE-19 cells, Bst2 knockdown did not significantly change SA- β -gal positivity, p53/p21/p16 expression, or γ -H2AX foci compared with shScramble controls (**new Fig. 2**). We confirmed these results at the in vivo level. Following Dox subretinal injection, SA- β -gal activity was quantified in the RPE flat mounts, showing 45.1% (shScramble) and 50.9% (shBst2) SA- β -gal-positive areas, respectively (**new Fig. 3c**), with no significant difference between the Dox-injected groups. RPE flat mount immunofluorescence further showed that p16 as well as p53/p21 was upregulated by Dox, but the expression did not differ between shScramble and shBst2 (**new Fig. 3a**). During the revision, we also evaluated electroretinography (ERG). The c-wave amplitude (reflecting RPE function) was similarly reduced in both Dox-injected shScramble and shBst2 mice compared with vehicle controls (**new Fig. 3d**), and a-wave and b-wave amplitudes were likewise reduced without differences between shScramble and shBst2 (**new Fig. 3d**). Collectively, these complementary in vitro and in vivo data indicate that Bst2 knockdown does not measurably alter canonical senescence phenotypes or RPE/retinal functional measures in our models. We have revised the Results section accordingly (**lines 170-189, page 7**) and included these new data in revised **Figs 2 and 3**.

Comment 3: There isn't adequate data showing co-localization of Bst2 and classic senescent cell markers, in vitro or in vivo. For example, in Fig2h, Bst2 and p53 are completely separated from each other in the RPE layer.

Response: We thank the reviewer for this valuable observation. We agree that the previous co-staining in Fig. 2h (**now Fig. 1h** in the revised manuscript) was suboptimal and could give the impression that Bst2 and p53 signals were spatially separated. To address this concern, we repeated the co-localization experiments using improved antibodies and imaging conditions, and we also performed an additional preparation optimized for assessing co-localization.

First, using archived young and aged mouse cryosections, we re-stained the RPE/retina with validated rabbit monoclonal antibodies against p53 (Cell Signaling #2527) and p21 (Cell Signaling #37543), and acquired images at higher magnification and resolution. In the updated images (**new Fig. 1h**), p53 and p21 signals localize to RPE nuclei, enabling a clearer assessment of marker distribution. Second, to more rigorously evaluate co-localization at the cell level, we newly prepared young and aged mouse RPE flat mounts and performed Bst2 co-staining with p53, followed by quantification of Bst2⁺/p53⁺ double-positive cells. These data directly demonstrate that Bst2 upregulation occurs in RPE cells that co-express canonical senescence markers, and we therefore incorporated these results into new **Fig. S1**. In addition, because co-localization after senolytic treatment is also central to the interpretation of our in vivo efficacy data, we performed additional co-staining experiments in the aged-mouse treatment group and updated the

corresponding figure (**Fig. 6**). Specifically, in retinal sections from 24-month-old mice treated with vehicle versus ABT-263@B-Z-PON, we demonstrate Bst2 co-localization with p53 and p21 in the RPE and quantify Bst2⁺p53⁺ and Bst2⁺p21⁺ cells, demonstrating a significant reduction after treatment (**Fig. 6d**).

Regarding the reviewer's specific point about the prior p53 signal pattern, the previously observed BM-like signal was attributable to antibody-related staining artifacts. For Bst2, while we note that this antibody can show a non-specific band-like signal near the BM region, the increased punctate Bst2 signal within the RPE cytoplasm is consistently observed in aged RPE and is the feature we interpret as biologically meaningful. To minimize potential autofluorescence, we used TrueBlack treatment and consistent acquisition settings in all groups. Together, these new experiments strengthen the evidence for Bst2 co-localization with classic senescence markers in vivo, and the revised manuscript has been updated accordingly (**updated Figs. 1 and 6**, with the flat mounts co-localization quantification added to **Fig. S1; lines 149-153 of page 5 and lines 335-336 of page 20**).

Comment 4: Bst2 seems to be expressed in all of the RPE cells in Fig2h, in aged mice.

Response: We thank the reviewer for this important observation. We agree that in the original Fig. 2h, Bst2 signal could appear broadly distributed in the aged RPE, potentially giving the impression that all aged RPE cells are Bst2-positive. As described in our response to Comment 3, we repeated the co-localization experiments and added flat mounts-based co-staining with quantitative analysis (**updated Figs. S1 and 6**). These new data show that Bst2 is increased in aged RPE but is not uniformly expressed across all RPE cells; rather, it is enriched in a subset of RPE cells that co-localize with canonical senescence markers (p53 and p21), as supported by quantification of Bst2⁺/p53⁺ and Bst2⁺/p21⁺ double-positive cells. We note that Bst2 immunostaining can show some background near the Bruch's membrane region; therefore, we base our interpretation primarily on cell-level co-localization and double-positive quantification, rather than Bst2 signal alone. We have updated the manuscript accordingly (**updated Figs. 1 and 6; new Fig.S1; lines 149-153 of page 5 and lines 335-336 of page 20**).

Comment 5: Bst2 is also highly expressed in non-RPE cells (IS/OS layer) in aged mice, in Fig2h.

Response: We appreciate the reviewer's comment. We acknowledge that in the original Fig.2h, signal was seen in the IS/OS region of aged sections. To address this, we repeated the staining and re-imaged the sections under optimized acquisition settings and included a lipofuscin/autofluorescence quenching step (TrueBlack). In the revised images (**updated Fig. 1h**), the apparent IS/OS-layer signal is largely eliminated, supporting that the prior IS/OS signal most likely reflected age-associated autofluorescence/background rather than specific Bst2 immunoreactivity. We have updated the Methods to state the use of TrueBlack for autofluorescence quenching (**lines 534-536, page 29**). For additional context, our ongoing internal analysis of retina scRNA-seq data indicates that Bst2 expression in rod and cone photoreceptors is very low and does not show a significant age-dependent increase (analysis ongoing; data not shown). These scRNA-seq observations are consistent with the revised immunostaining results and further support the conclusion that Bst2 enrichment in our study is primarily associated with senescent RPE cells. Accordingly, our conclusions regarding Bst2 as a practical targeting marker are based on RPE-localized Bst2 signal and its cell-level co-localization with senescence markers (p53/p21), as shown in the revised figures.

Comment 6: In Fig2h, p53, a nuclear protein, does not overlap with the nuclei.

Response: This issue was addressed in our response to Comment 3 above: we repeated the co-staining with validated rabbit monoclonal antibodies and improved imaging/acquisition settings, and the updated high-magnification confocal images now show clear nuclear localization of p53 (updated **Figs. 1 and 6; Fig. S1**).

Comment 7: Since this study aims at therapeutic development for treating AMD, is there evidence of high expression of *Bst2* in senescent RPE cells in human AMD patient samples?

Response: We thank the reviewer for this question. To assess the translational relevance of BST2 in human RPE, we analyzed three publicly available transcriptomic datasets (GSE135092, GSE135133, and GSE210543) in the revision. In GSE135092 (bulk RPE/choroid), BST2 expression increased significantly with donor age within both the AMD and normal groups (Fig. R2 and R3). In GSE135133 (snRNA-seq of 1,704 RPE nuclei from control donor eyes), BST2 expression increased significantly with age (Fig. R4). In GSE210543 (developmental/lifespan dataset), BST2 expression progressively increased from fetal to adult stages (Fig. R5). Although AMD-control differences were modest after adjusting for age, in each of these independent datasets, BST2 showed a consistent positive association with age in human RPE, supporting BST2 as an aging/senescence-associated marker and providing translational rationale for BST2-guided targeting. We have added a brief statement in the Discussion to highlight this age-associated increase in human datasets (**lines 399-401, page 25**) and have included the corresponding source publications for these datasets as new references (Refs. #51,52).

Figure R2. Age-BST2 expression correlation in AMD samples (GSE135092)

Figure R3. Age-BST2 expression correlation in normal aging samples (GSE135092)

Figure R4. snRNA-seq data of age-*BST2* expression correlation in control samples (GSE135133)

Figure R5. age-*BST2* expression correlation in normal and fetal samples (GSE210543)

Comment Section B. *Bst2*-targeted delivery to senescent RPE cells.

Comment 1: *ABT263@B-Z-PON is intravitreally injected, but there are retina and other layers of anatomical barriers between vitreous and the RPE layer. How could the bulky nanoparticle with several components including antibody cross these layers and end up being uptaken by RPE cells?*

Response: We thank the reviewer for raising this important question. Previous studies have demonstrated that nanoparticles administered via intravitreal injection can traverse retinal layers and reach the RPE, and nanoparticle-based intravitreal delivery is increasingly recognized as a feasible strategy for retinal therapy (*Rajala et al. Nano letters, 2014 / Adjianto et al. European Journal of Pharmaceutics and Biopharmaceutics, 2015 / Nardella et al. Biomolecules, 2025 / Li et al. European Journal of Pharmaceutical Sciences, 2026 / Li et al. Journal of Controlled Release, 2026*). Our nanoplatform is based on ultrasmall mesoporous silica nanoparticles (~30-40 nm), which provide a compact and biocompatible scaffold for drug loading and surface functionalization, a property favorable for vitreous diffusion and trans-retinal transport compared with bulkier carriers. Although the hydrodynamic size of the full antibody-conjugated formulation (*ABT-263@B-Z-PON*) was not directly measured in the current study (Fig. S5), antibody conjugation is expected to induce only a modest increase in particle size. Consistent with this, our previous work using the same nanoparticle scaffold showed a rightward shift in DLS profiles upon conjugation with an IgG-sized protein, while remaining within the nanoscale range (Ref. #37, *Oh et al., Chem. Eng. J., 2023, Fig. 3g, reproduced below for the reviewer's reference*).

Importantly, and most directly addressing this concern, we performed additional in vivo experiments using fluorescently labeled B-Z-PONs. RPE flat mounts imaging clearly demonstrated localization of B-Z-PON signals within the RPE following intravitreal injection in Dox-induced mouse models (updated **Fig. 5b**). Together, prior reports and our direct imaging data support the feasibility of retinal penetration and RPE uptake of *ABT-263@B-Z-PON* following intravitreal administration. We added this point in the Results section (**lines 293-299, page 16**).

Oh et al., Chem. Eng. J., 2023, Fig. 3g.

Comment 2: Would microglia and other immune cells clear most of the ABT263@B-Z-PON particles?

Response: We appreciate the reviewer's thoughtful question. We agree that clearance by microglia and other ocular immune cells is an important consideration for intravitreally delivered nanocarriers. Although immune uptake cannot be completely excluded, our platform was designed to mitigate nonspecific interactions and premature clearance. In our previous work using the same plug-and-play nanoparticle scaffold (Ref. #37), protein-based surface functionalization reduced nonspecific protein adsorption and attenuated immune-cell uptake, consistent with anti-fouling behavior. In the present study, B-Z-PON adopts a similar design principle and uses an Fc-anchored antibody orientation strategy, which can reduce exposed Fc availability and may help limit Fc receptor-mediated recognition compared with randomly adsorbed antibodies. Importantly, fluorescently labeled B-Z-PON showed robust accumulation in the RPE after intravitreal injection (**updated Fig. 5b**), indicating that immune clearance did not prevent effective RPE targeting in this model.

Comment 3: Please provide the evidence that ABT263@B-Z-PON is indeed uptaken by RPE cells of Dox-treated mice and 24-month old mice after intravitreal injection. It is hard to tell from Fig5b whether it is uptaken by senescent RPE cells.

Response: We thank the reviewer for this important comment. We agree that direct evidence of RPE uptake of ABT-263@B-Z-PON is critical for interpreting the in vivo efficacy. To directly visualize RPE uptake, we performed intravitreal injection of DiD-labeled Z-PON (DiD@Z-PON) and DiD@B-Z-PON in the Dox-induced RPE model. DiD@B-Z-PON showed robust accumulation in the RPE layer and preferential co-localization with p21-positive senescent RPE cells, whereas DiD@Z-PON showed minimal RPE uptake (**updated Fig. 5b**). These data provide direct evidence of senescent RPE-selective uptake in the Dox-treated model. We acknowledge that DiD-labeling experiments were not performed in naturally aged (24-month-old) mice. However, in this aging model, we provide complementary in vivo evidence supporting effective RPE targeting through therapeutic and histological outcomes. Specifically, ABT-263@B-Z-PON treatment in 24-month-old mice led to a marked reduction of Bst2⁺/p53⁺ and Bst2⁺/p21⁺ senescent RPE cells, as demonstrated by high-resolution immunofluorescence analysis (**new Fig. 6d**). The coordinated loss of Bst2 and canonical senescence markers in the RPE strongly supports effective

delivery and intracellular action of ABT-263@B-Z-PON in aged RPE. Because these data are central to addressing the reviewer's concern, we have moved the co-localization and elimination data in aged mice to the main figures (**new Fig. 6d**) and clarified the interpretation in the revised Results and Discussion sections (**lines 293-299 of page 16 and lines 335-336 of page 20**).

Comment 4: What is the evidence that GSH is drastically higher in senescent RPE cells than normal RPE cells?

Response: We thank the reviewer for this important point. In our Dox-induced senescent ARPE-19 model, we directly measured intracellular glutathione (GSH) and observed an approximately threefold increase compared with non-senescent controls (Fig. S3). However, we agree that elevated GSH is not specific to senescence and can also occur under diverse stress conditions, including Dox exposure. We have therefore revised the manuscript to avoid implying that redox responsiveness alone confers senescence specificity. Instead, the redox-responsive component is intended to facilitate efficient intracellular drug release after nanoparticles are internalized, whereas cell-type/phenotype selectivity is primarily provided by Bst2 antibody-mediated targeting. As noted above, altered GSH/redox homeostasis has also been reported in certain senescence models (Ref. #41). Finally, we now briefly acknowledge the limitation that stressed but non-senescent cells may also show increased GSH; nevertheless, in our platform, uptake is restricted by Bst2 targeting and redox responsiveness functions as an intracellular release mechanism rather than a senescence marker. We revised the manuscript accordingly (**lines 193-196, page 9**).

Comment 5: Please explain how ABT263@B-Z-PON can escape from endosomes?

Response: We thank the reviewer for this important question. Following cellular uptake, ABT-263@B-Z-PON is expected to enter cells predominantly via endocytic pathways and traffic through endosomal and lysosomal compartments. While precise molecular mechanisms of endosomal escape are difficult to define for most nanocarriers, our platform is not designed to induce active membrane disruption or physical endosomal rupture. Instead, it exploits the biochemical environment of endo/lysosomal compartments to enable effective intracellular drug release. Specifically, the mesoporous organosilica scaffold of ABT-263@B-Z-PON is engineered to undergo progressive destabilization under reductive conditions, such as those encountered in GSH-rich intracellular compartments. Consistent with this design principle, TEM analysis demonstrated accelerated structural degradation of the carrier under reducing conditions (Fig. S4). Importantly, ABT-263 is a small-molecule compound capable of diffusing across intracellular membranes after release, allowing access to its cytosolic targets even when liberation occurs within endo/lysosomal compartments. Together, these features indicate that ABT-263@B-Z-PON functionally overcomes endosomal confinement through environment-triggered carrier disassembly and intracellular drug release, rather than through active membrane disruption. This mechanism is described in the manuscript (**lines 199–211, page 9**).

Comment 6: In Figs 5 and 6, ABT263@B-Z-PON improves ERG a, b, and c waves. Is the retina also morphologically improved? e.g., the outer nuclear layer thickness?

Response: We thank the reviewer for this important suggestion. To evaluate whether the ERG improvements are accompanied by structural rescue, we performed additional histological analyses in both in vivo models. We added H&E-stained retinal sections and quantified outer nuclear layer (ONL) thickness in (i) Dox-induced senescent RPE mice and (ii) naturally aged (24-

month-old) mice. In both models, ABT263@B-Z-PON treatment significantly increased/preserved ONL thickness compared with vehicle controls, indicating morphological improvement consistent with the functional recovery observed by ERG. These new histology images and ONL quantification have been incorporated into the revised manuscript as part of the updated main figures (**revised Figs. 5 and 6**), and the text has been revised accordingly (**lines 307-309 of page 16 and lines 341-343 of page 20**).

Comment 7: Fig6e, in 24-month old mice, RPE cell division should have long ended. Please explain why ABT263@B-Z-PON leads to cell division as shown by Ki67-marked double nuclei in each RPE cell.

Response: We thank the reviewer for this insightful and important comment. We agree that canonical mitotic proliferation of RPE cells is not expected in 24-month-old mice, and the reviewer's concern is well founded. Importantly, we do not interpret the Ki67-positive, binucleated RPE profiles observed after ABT263@B-Z-PON treatment as evidence of widespread or pathological RPE proliferation. Rather, binucleation in long-lived epithelial tissues, including the RPE, is commonly associated with limited, adaptive cell-cycle-associated responses such as incomplete cytokinesis or polyploidization, rather than true cell division. Consistent with this view, previous studies have reported Ki67-positive RPE cells even in aged human donor eyes, indicating that aging does not completely abolish the capacity for cell-cycle re-entry in RPE cells (added Ref. #42). Moreover, Ki67 expression has been preferentially observed in binucleated RPE cells in contexts of tissue stress or remodeling (added Ref. #43). Notably, in our study, Ki67-positive binucleated RPE cells were observed in parallel with clear functional and structural improvement, including enhanced ERG a-, b-, and c-wave responses, restoration of RPE junctional integrity (ZO-1), increased ONL thickness, and marked reduction of senescence markers (p53, p21, and SA- β -gal). Similar transient cell-cycle-associated responses following senolytic intervention in aged RPE have also been reported in our prior studies (Refs. #21, 39). Together, these findings support the interpretation that the observed Ki67 signal reflects a restricted, adaptive response following senescent cell clearance, compatible with functional recovery rather than aberrant proliferation. We have clarified this interpretation in the revised Results section (**lines 337-341, page 20**).